# DATA GEOMETRY DEPENDENT BOUNDS ON NETWORK WIDTHS IN DEEP RELU NETWORKS

## ABSTRACT

While existing theoretical and empirical findings in neural networks indicate that complex datasets often necessitate expansive network architectures for optimal classification, there remains a gap in precisely determining the specific network structure required for complete dataset classification. This paper addresses this gap by establishing bounds on ReLU network widths based on the geometric characteristics of the dataset. Specifically, we propose neural network architectures suitable for scenarios where the dataset can be effectively separated by a collection of polytopes. We also provide both theoretical and empirical evidences that gradient descent converges to the proposed network configurations. Lastly, we propose an algorithm that finds such a polytope cover for the given dataset, which empirically demonstrates that every class of MNIST, Fashion-MNIST, and CIFAR10 can be distinguished by at most three polytopes.

## 1 INTRODUCTION

To understand the high performance of deep neural networks (DNNs), there have been numerous studies investigating their capacity, generalization, memorization, and universal approximation property (UAP). Since Cybenko (1989) first proved UAP of two-layer neural network on a compact set, UAP of DNNs has been extensively investigated in various settings. The minimal depths and widths of deep ReLU networks to have UAP have been studied (Hornik, 1991; Park et al., 2020). Furthermore, the number of parameters required to memorize given arbitrary points was also studied (Yun et al., 2019; Bubeck et al., 2020). Investigating the complexity of neural networks in terms of the number of linear partitions (Serra et al., 2018; Hanin & Rolnick, 2019) or the bound on the Betti numbers (Bianchini & Scarselli, 2014) is another way of studying the representation power of the neural networks. These fundamental results on neural networks help us to understand the relationship between approximation power and neural network architectures.

On the other hand, the effect of training dataset characteristics on the training network architectures in terms of UAP has not been explicitly solved. For instance, for the swiss roll dataset given in Figure 1(a), *what is the required depth and width to perfectly classify this dataset?* While this is a

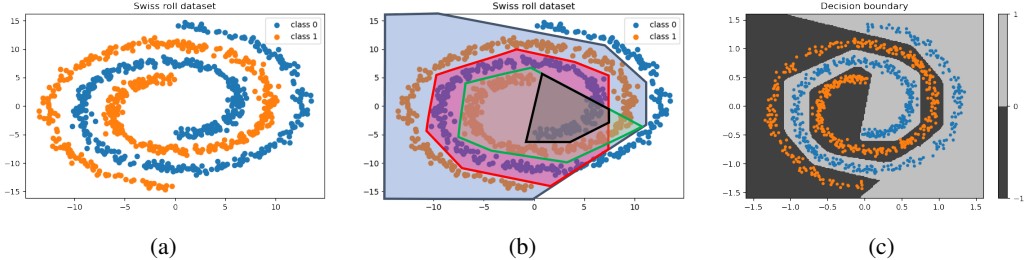

(a)        (b)        (c)

Figure 1: What type of neural network architecture is capable of effectively classifying the Swiss roll dataset depicted in (a)? By establishing a set of covering polytopes to enclose one class, as illustrated in (b), our result demonstrates that a three-layer ReLU network with the architecture $2 \xrightarrow{\sigma} 24 \xrightarrow{\sigma} 4 \rightarrow 1$ can successfully achieve this classification task, as exemplified in (c).

practical question for training neural networks, existing theoretical results on UAP (Hornik, 1991; Park et al., 2020) only provide trivial lower bounds (depth 2 and width 3, in this case). While empirical observations suggest that increasing both depth and width can eventually lead to success, there is no theoretical insight available to predict this outcome.

In this paper, we address this problem with the geometric perspective on deep ReLU networks in terms of the convex polytope structure. This was inspired from many recent geometric results on DNNs (Black et al., 2022; Grigsby & Lindsey, 2022; Berzins, 2023; Huchette et al., 2023). Specifically, we provide lower and upper bounds of network depth and widths in terms of its polytope cover to completely classify the given data class. For example, our result in Theorem 3.4 clarifies that the swiss roll dataset in Figure 1 can be classified by a three-layer ReLU network with 28 neurons.

Theoretically, our goal is to delve into the following fundamental approximation problem: *for a given topological space $\mathcal{X}$ representing the dataset and $\varepsilon > 0$, what is an upper bound on the widths of a neural network $\mathcal{N}$ such that $\mathcal{N}(\boldsymbol{x}) = 1$ for $\boldsymbol{x} \in \mathcal{X}$, and it vanishes outside the $\varepsilon$ neighborhood of $\mathcal{X}$?* To answer this fundamental question, we explicitly construct a neural network based on a polytope cover of the given data manifold $\mathcal{X}$, which relies on the geometric characteristics.

Our contributions can be summarized as follows.

- For a convex polytope $\mathcal{X} \subset \mathbb{R}^d$, we develop a three-layer ReLU network that can approximate the indicator function of $\mathcal{X}$ under a given $\varepsilon$ error. We provide upper and lower bounds of width in terms of the number of faces of the convex polytope (Proposition 3.2). We propose the concept of *polytope-basis cover* (Definition 3.3) and we refine our result for a four-layer ReLU network when $\mathcal{X}$ has a polytope-basis cover (Theorem 3.4).

- Applying our results, we derive upper bounds of network widths when $\mathcal{X}$ is a simplicial complex, or can be covered by a difference of polytopes-shaped holes (Theorem 3.5 and 3.6). Our terms depend on the dimension $d$, and the number of facets, or the Betti numbers of $\mathcal{X}$. This result describe how the bounds vary according to geometric complexity of $\mathcal{X}$.

- For the cross-entropy and the squared loss, under specific assumptions on the dataset and initialization conditions, we prove the existence of a discrete path that strictly decreases to the zero loss (Theorem 4.3). This suggests that our proposed neural networks can be achieved by gradient descent method, which is also verified by experiments.

- In a practical application of our theoretical findings, we propose a method to investigate the geometric characteristics of the given dataset by training a two-layer ReLU network (Proposition 3.7). The empirical results show that every class in MNIST, Fashion-MNIST, and CIFAR10 datasets can be separated by at most three convex polytopes, which reveals simple geometric characteristics of the real datasets (Table 1 in Appendix E).

## 2 RELATED WORK

**Geometrical approach for deep ReLU networks.** Various geometric approaches have been employed to investigate the approximation capabilities of deep ReLU networks. For instance, Hanin & Rolnick (2019) introduced the concept of bent hyperplanes in the input space, measuring its complexity both theoretically and empirically. This approach has found applications in diverse research areas, including the analysis of decision regions (Beise et al., 2021; Grigsby & Lindsey, 2022; Black et al., 2022) and the characterization of linear regions within ReLU networks (Rolnick & Kording, 2020).

In recent years, there has been a growing interest in investigating polytope structures induced by deep ReLU networks (Fawzi et al., 2018; Xu et al., 2021; Vincent & Schwager, 2022; Black et al., 2022; Haase et al., 2023; Liu et al., 2023; Fan et al., 2023; Huchette et al., 2023; Vallin et al., 2023). Fawzi et al. (2018) focused on analyzing the geometrical features of the decision boundary of trained networks. Similarly, Masden (2022) introduced algorithms capable of extracting the polytope structure inherent in networks and deriving topological properties of the decision boundary. Carlsson (2019) and Vallin et al. (2023) considered the pre-image of ReLU networks, characterizing the geometric shapes of the decision boundary to gain an understanding of the polytope partitions of deep ReLU networks.

However, there has been limited exploration into how neural networks can be explicitly constructed to classify a given dataset, as depicted in Figure 1. In this work, we address the problem of approximating the indicator function defined on polytopes in terms of network architecture. Indeed, approximating the indicator function is generally more difficult than just designing a classifier. This is because if a neural network can effectively approximate the indicator function on a given dataset class, it inherently possesses the capability to classify $\mathcal{X}$ from other classes, but it does not hold in the reverse direction. We also mention that approximating the indicator function is a sufficient condition to achieve UAP (Proposition C.1, Theorem C.5). Therefore, we employ a similar argument on arranging bent hyperplanes to derive bounds on network widths dependent on the geometric features of the given dataset to approximate the indicator function. This approach can be viewed as a converse method for studying the approximation ability of neural networks, a topic that has not been extensively explored.

**Dataset geometry and neural networks.** Several studies have explored the relationship between the geometric features of datasets and neural network training, often assuming low-dimensional properties of the data manifold (Buchanan et al., 2020; Wang et al., 2021; Chen et al., 2022; Tiwari & Konidaris, 2022). For example, Buchanan et al. (2020) and Wang et al. (2021) addressed the task of distinguishing between two curves, investigating the convergence speed and generalization concerning the geometric features of the dataset. Similarly, for low-dimensional data manifolds, Tiwari & Konidaris (2022) examined the effects of data geometry on the complexity of trained neural networks by measuring the distance to the manifold. In a similar vein, Dirksen et al. (2022) considered the separation problem with random ReLU networks, and provided a lower bound of widths in terms of geometric property of datasets. It is worth noting that they proposed a concept of mutual covering of data points, akin to our polytope covering on each class. Additionally, there are many empirical studies that also support the implicit relationship between network architecture and geometric complexity or topological structure of the data manifold (Fawzi et al., 2018; Kim et al., 2020; Cohen et al., 2020; Birdal et al., 2021; Barannikov et al., 2021a;b; Magai & Ayzenberg, 2022; Tiwari & Konidaris, 2022). However, still there exists a gap in the literature when it comes to explicitly constructing a neural network that precisely fits the given data manifold. This paper aims to bridge this gap by exploring the explicit relationship between data geometry and neural network architecture, focusing on the polytope structure induced by ReLU.

## 3 DATA-GEOMETRY AND NEURAL NETWORK ARCHITECTURE

### 3.1 PRELIMINARIES

**Notation.** Throughout the paper, we denote scalars by lowercase letters and vectors by boldface lowercase letters. For a positive integer $m$, $[m]$ represents the set $\{1, 2, \cdots, m\}$. The ReLU activation function is denoted by $\sigma(x) := \text{ReLU}(x) = \max\{0, x\}$, and it is applied to a vector coordinate-wisely. The sigmoid activation function is denoted as $\text{SIG}(x) = \frac{1}{1+e^{-x}}$. The max pooling operation is represented as $\text{MAX} : \mathbb{R}^d \to \mathbb{R}$, which returns the maximum component of the input vector. The $\varepsilon$ neighborhood of a topological space $\mathcal{X} \subset \mathbb{R}^d$ is defined by $\mathcal{B}_\varepsilon(\mathcal{X}) := \{\boldsymbol{x} \in \mathbb{R}^d : \min_{\boldsymbol{y} \in \mathcal{X}} \|\boldsymbol{x} - \boldsymbol{y}\|_2 < \varepsilon\}$. For a topological space $\mathcal{X}$, the indicator function over $\mathcal{X}$ is denoted by

$$\mathbb{1}_{\{\mathcal{X}\}}(\boldsymbol{x}) := \begin{cases} 1, & \text{if } \boldsymbol{x} \in \mathcal{X}, \\ 0, & \text{otherwise.} \end{cases}$$

**Network architectures.** To denote the neural network architectures (depth and hidden layer widths), we use the following notation. A $k$-layer neural network $\mathcal{N} : \mathbb{R}^d \to \mathbb{R}$ with hidden layer widths $d_1, d_2, \cdots, d_{k-1}$ and activation functions $\text{ACT}_1, \text{ACT}_2, \cdots, \text{ACT}_k$ is represented by $d \overset{\text{ACT}_1}{\to} d_1 \overset{\text{ACT}_2}{\to} d_2 \overset{\text{ACT}_3}{\to} \cdots \overset{\text{ACT}_{k-1}}{\to} d_{k-1} \overset{\text{ACT}_k}{\to} 1$. When the activation function is the identity, we add nothing on the arrow. In this paper, the terminology *architecture* refers the structure of the neural network, which means the depth and the width of hidden layers. For example, $d \overset{\sigma}{\to} l \to 1$ denotes a two-layer ReLU network with $l$ neurons, presented by

$$\mathcal{N}(\boldsymbol{x}) = v_0 + \sum_{k=1}^{l} v_k \sigma(\boldsymbol{w}_k^\top \boldsymbol{x} + b_k). \tag{1}$$

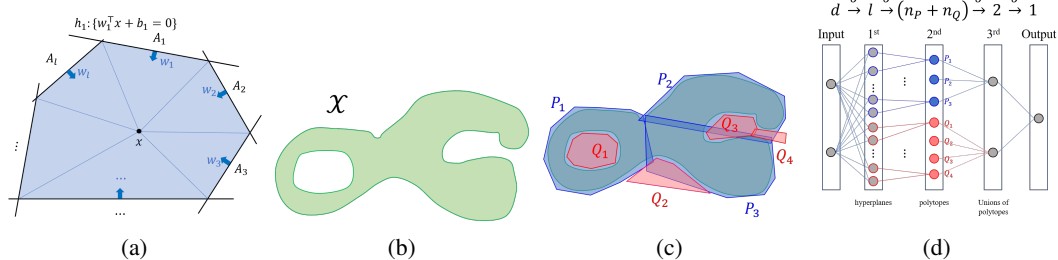

Figure 2: **The fundamental idea of our work. (a)** A convex polytope enclosed by $l$ hyperplanes can be decomposed by $l$ small pyramids. **(b)** A is a topological space $\mathcal{X} \subset \mathbb{R}^d$ is given. **(c)** A collection of polytopes $\mathcal{C} = \{P_1, P_2, P_3, Q_1, Q_2, Q_3, Q_4\}$ is a polytope-basis cover of $\mathcal{X}$. **(d)** A feasible architecture on $\mathcal{X}$ with margin $\varepsilon$, described by the polytope-basis cover presented in (c). The constructive proof in Theorem 3.4 further exhibits the role of neurons in hidden layers: hyperplanes, convex polytopes, and union of polytopes.

The architecture of networks is often referred to as $\mathcal{A}$. We further introduce a novel terminology on network architectures: a *feasible architecture* of networks on a topological space $\mathcal{X}$.

**Definition 3.1.** *Let $\mathcal{X} \subset \mathbb{R}^d$ be a topological space and $\varepsilon > 0$ be a given margin. A neural network architecture $\mathcal{A}$ is called a* feasible architecture on $\mathcal{X}$ with margin $\varepsilon$ *if there exists a neural network with the architecture $\mathcal{A}$ such that $\mathcal{N}(\mathbb{R}^d) \subset [0, 1]$ and*

$$\mathcal{N}(\boldsymbol{x}) = 1 \quad \text{if} \quad \boldsymbol{x} \in \mathcal{X},$$
$$\mathcal{N}(\boldsymbol{x}) = 0 \quad \text{if} \quad \boldsymbol{x} \notin \mathcal{B}_\varepsilon(\mathcal{X}).$$

*If this property can hold on every $\varepsilon > 0$, then we simply say that $\mathcal{A}$ is a* feasible architecture on $\mathcal{X}$.

In other words, it refers to the architecture of neural networks that can approximates the indicator function on the given manifold $\mathcal{X}$, and be vanished for inputs farther than $\varepsilon$ from $\mathcal{X}$ [1]. The objective of this paper is to investigate the relationship between feasible architectures and geometrical complexity of the dataset.

### 3.2 MAIN THEORETICAL FINDINGS

Let $\mathcal{X} \subset \mathbb{R}^d$ be a convex polytope with $l$ faces. Our first goal is to derive the upper and lower bounds of widths of a ReLU neural network to be a feasible architecture on $\mathcal{X}$. Applying piecewise linearity of ReLU networks and volume formula of convex polytopes, the following proposition provides upper and lower bounds of network widths to be a feasible architecture on $\mathcal{X}$.

**Proposition 3.2.** *Let $\mathcal{X} \subset \mathbb{R}^d$ be a convex polytope enclosed by $l$ hyperplanes. Then $d \xrightarrow{\sigma} l \xrightarrow{\sigma} 1$ is a feasible architecture on $\mathcal{X}$ with minimal depth. Conversely, if $d \xrightarrow{\sigma} d_1 \xrightarrow{\sigma} d_2 \xrightarrow{\sigma} \cdots \xrightarrow{\sigma} d_k \xrightarrow{\sigma} 1$ is a feasible architecture on $\mathcal{X}$, then*

$$d_1 \cdot \prod_{j=2}^{k} (2d_j + 1) \geq \begin{cases} \left\lceil \frac{l}{2} \right\rceil + (d-2), & \text{if} \quad l \geq 2d+1, \\ 2d-1, & \text{if} \quad l = 2d-1, 2d, \\ d+1, & \text{if} \quad l < 2d-1. \end{cases}$$

*This lower bound is optimal when $k = 1$ and $d = 2$.*

*Proof sketch.* We briefly introduce the main idea here. Let $C$ be the enclosing convex polytope, where $A_1, \cdots, A_l$ are enclosing hyperplanes. Let $\boldsymbol{x}$ be a point in $C$. Since $C$ is convex, it can be decomposed to $l$ pyramids whose common apex is $\boldsymbol{x}$ (see Figure 2(a)). Then, the volume (in Lebesgue sense) of $\mathcal{X}$ is equal with the sum of $l$ pyramids. Mathematically, it becomes

$$\text{Vol}_d(C) = \frac{1}{d} \sum_{i=1}^{l} \text{Vol}_{d-1}(A_i) \sigma(\boldsymbol{w}_i^\top \boldsymbol{x} + b_i)$$

---

[1] Furthermore, it is worth noting that the result concerning the approximation of the indicator function directly leads to the UAP result, as outlined in Theorem C.5).

where $\boldsymbol{w}_i$ is a unit vector of the hyperplane $A_i$, and $\mathrm{Vol}_d$ denotes the $(d-1)$-dimensional volume. From this equation, we define a two-layer ReLU network $\mathcal{N}(\boldsymbol{x}) := \mathrm{Vol}_d(C) - \frac{1}{d}\sum_{i=1}^{l}\mathrm{Vol}_{d-1}(A_i)\sigma(\boldsymbol{w}_i^\top \boldsymbol{x} + b_i)$. Then $\mathcal{N}(\boldsymbol{x})$ has a constant output in $C$, and we can prove that $d \xrightarrow{\sigma} l \xrightarrow{\sigma} 1$ is a feasible architecture on the polytope, by adjusting some coefficients of $\mathcal{N}$. The detail proof can be found in Appendix B.1. □

The proof of Proposition 3.2 suggests how ReLU networks can approximate the indicator function on a convex polytope. Building upon this proposition, we extend our findings to arbitrary topological spaces, specifically those that can be tightly covered by a collection of polytopes. To facilitate this extension, we introduce an additional terminology.

**Definition 3.3.** *For a given topological space $\mathcal{X} \subset \mathbb{R}^d$ and $\varepsilon > 0$, a finite collection of polytopes $\mathcal{C} := \{P_1, \cdots, P_{n_P}, Q_1, \cdots, Q_{n_Q}\}$ is called a* polytope-basis cover *of $\mathcal{X}$ with margin $\varepsilon$ if the set difference $D := \bigcup_{i \in [n_P]} P_i - \bigcup_{j \in [n_Q]} Q_j$ satisfies $\mathcal{X} \subset D \subset \mathcal{B}_\varepsilon(\mathcal{X})$.*

A polytope-basis cover of $\mathcal{X}$ is an approximation of manifold $\mathcal{X}$ within the margin $\varepsilon$, consisting of convex polytopes, using set operations union and difference. Figure 2(d) shows one example of a polytope-basis cover of $\mathcal{X}$ given in Figure 2(c). Then, we can provide a feasible architecture on $\mathcal{X}$ in terms of its polytope-basis cover, using the upper bound proposed in Proposition 3.2.

**Theorem 3.4.** *For a given topological space $\mathcal{X} \subset \mathbb{R}^d$ and $\varepsilon > 0$, let $\mathcal{C} = \{P_1, \cdots, P_{n_P}, Q_1, \cdots, Q_{n_Q}\}$ be a polytope-basis cover of $\mathcal{X}$ with margin $\varepsilon$. Let $l$ denote the total number of faces of the convex polytopes in $\mathcal{C}$. Then, $d \xrightarrow{\sigma} l \xrightarrow{\sigma} (n_P + n_Q) \xrightarrow{\sigma} 2 \xrightarrow{\sigma} 1$ is a feasible architecture on $\mathcal{X}$ with margin $\varepsilon$.*

The proof can be found in Section B.2 in Appendix. One of the important contributions of Theorem 3.4 is that its construction exhibits the exact role of each neuron in the hidden layers. See a polytope-basis cover represented in Figure 2(c). Each neuron in the first hidden layer represents a hyperplane in the input space $\mathbb{R}^d$, where each neuron in the second hidden layer represents a convex polytope ($P_i$ or $Q_j$) in $\mathcal{C}$ that is formed by connected neurons in the first layer. Similarly, two neurons in the third hidden layer represent two groups of polytopes that constitutes the polytope-basis cover. This geometric insight offers an interpretation for the concept of high-level polytopes introduced by Xu et al. (2021).

On the other hand, a simplicial $m$-complex is a type of simplicial complex where the highest dimension of any simplex equals $m$. For a given simplicial complex $K$, a facet of $K$ is a maximal simplex which does not serve as a face of any larger simplex. Here, the Betti number is a key metric used in TDA to denote the number of $k$-dimensional 'holes' in a data distribution, which are frequently employed to study the topological characteristics of topological spaces. With these definitions in mind, in the following Theorem 3.5, we first derive a feasible architecture on a simplicial complex $\mathcal{X}$. Specifically, Theorem 3.5 provides a general network architecture depends on the geometric structure of the dataset $\mathcal{X}$, especially, on a polytope-basis cover of it. In essence, if there exists prior information on the dataset geometry, this theorem establishes an upper bound on the necessary width and depth of feasible architectures. This result can be further tailored for datasets with specific structures, such as a simplicial complex.

**Theorem 3.5.** *Let $\mathcal{X} \subset \mathbb{R}^d$ be a simplicial $m$-complex consists of $k$ facets, and let $k_j$ be the number of $j$-dimensional facets of $\mathcal{X}$. Then, $d \xrightarrow{\sigma} d_1 \xrightarrow{\sigma} k \xrightarrow{MAX} 1$ is a feasible architecture on $\mathcal{X}$, where $d_1$ is bounded by*

$$d_1 \leq \min\left\{ k(d+1) - (d-1)\left\lfloor \sum_{j=0}^{\lfloor \frac{d-1}{2} \rfloor} \frac{k_j}{2} \right\rfloor,\ (d+1)\left[ \sum_{j \leq \frac{d}{2}} \left( k_j \frac{j+2}{d-j} + \frac{j+2}{j+1} \right) + \sum_{j > \frac{d}{2}} k_j \right] \right\}. \quad (2)$$

The proof of this theorem can be found in Section B.3. Theorem 3.5 reveals that the width $d_1$ is bounded by in terms of the dimension $m$ and the number of facets $k$ of the provided simplicial complex. Looking at this from a topological perspective, it is generally intuitive that a smaller number of facets suggests a simpler structure of the simplicial complex. This notion is mathematically expressed in (2), which suggests that when $m$ is fixed, the first maximum value in (2) results in $d_1 \lesssim \frac{k}{2}(d+3)$, which magnifies as $k$ increases. Similarly, when $m < \frac{d}{2}$ and $k$ is fixed, the summation

in the second maximum value in (2) reduces to $d_1 \lesssim (d+1)\left(k\frac{m+2}{d-m}+2\right)$, which rapidly diminishes as $m$ decreases. This analysis demonstrates that a smaller dimension $m$ demands smaller widths, which aligns with the intuition that a low-dimensional manifold could be approximated with fewer neurons.

The result in Theorem 3.4 can be also leveraged to ascertain a neural network architecture with width bounds defined in terms of the Betti numbers. Recall that the theorem offers an upper bound on widths when $\mathcal{X}$ can be depicted as a difference between groups of convex sets. Expanding on this, when $\mathcal{X}$ contains 'convex-shaped holes,' we can derive a bound of network architecture in relation to its Betti numbers. This concept is further explained in the following theorem.

**Theorem 3.6.** *Let $\mathcal{X}$ be a topological space obtained by removing some disjoint prism-shaped convex polytopes from a convex polytope. Let $l$ be the maximum number of faces of these polytopes. Let $\beta_k$ be the $k$-th Betti number of $\mathcal{X}$. Then,*

$$d \xrightarrow{\sigma} \left(l + 2(\beta_0 - 1) + \sum_{k=1}^{d}\left(l - 2(d - k - 1)\right)\beta_k\right) \xrightarrow{\sigma} \left(\sum_{k=0}^{d}\beta_k\right) \xrightarrow{\sigma} 2 \xrightarrow{\sigma} 1 \tag{3}$$

*is a feasible architecture on $\mathcal{X}$. Conversely, for any such $\mathcal{X}$, suppose $d \xrightarrow{\sigma} d_1 \xrightarrow{\sigma} d_2 \xrightarrow{\sigma} \cdots \xrightarrow{\sigma} d_k \xrightarrow{f} 1$ is a feasible architecture on $\mathcal{X}$ where the last activation function $f$ is either $\sigma$ or $\mathtt{SIG}$. Then, the network widths should satisfy*

$$\sum_{i=1}^{k}\prod_{j=i}^{k} d_j \geq 2\sum_{k=0}^{d}\beta_k - 2. \tag{4}$$

The proof is written in Appendix B.4. Theorem 3.6 introduces upper and lower bounds on network widths in terms of the Betti numbers of $\mathcal{X}$, connecting the topological characteristics of a dataset with upper bounds on network widths. We also show in Proposition C.4 that topological property alone cannot determine the feasible architecture, which demonstrates the significance of Theorem 3.6. Note that the result in Proposition C.4 also implies that the geometrical assumptions in the theorem is indispensable.

Interestingly, the sum of Betti numbers $\sum_{k=0}^{d}\beta_k$ in (3), which appears in the third layer, is often called the *topological complexity* of $\mathcal{X}$. This quantity is recognized as a measure of the complexity of a given topological space (Bianchini & Scarselli, 2014; Naitzat et al., 2020). This value has connections with other fields: for example, it has some lower and upper bounds from Morse theory (Milnor et al., 1963) and Gromov's Betti number Theorem (Gromov, 1981). On the other hand, the lower bound on widths shows that the sum of product of widths should be greater than the sum of Betti numbers. It also verifies that the contribution of the width in deeper layers holds greater significance compared to previous layers. This is the first result of completely characterizing neural network architecture having UAP in terms of topological dataset characteristics.

Lastly, we also point out that our findings in this section can be easily extended to other neural network architectures. In Appendix A, we broaden our results to encompass deep ReLU networks (Corollary A.1) and sigmoid activation function (Corollary A.2).

## 3.3 ANALYZING GEOMETRIC STRUCTURE OF REAL DATASET

So far, we have demonstrated how feasible architecture on $\mathcal{X}$ can be reduced from the geometric characteristics of $\mathcal{X}$. In this section, we explore the reverse scenario: given a neural network that achieves zero error on a finite dataset $\mathcal{D}$, can we extract geometric information about $\mathcal{D}$? Below, we address this question by presenting a method to analyze dataset geometry through a trained neural network. This method involves establishing a polytope-basis cover of the real dataset, which was assumed to be provided in Section 3.

**Proposition 3.7.** *Let $\mathcal{N}$ be a two-layer ReLU network with $l$ neurons defined by (1), where the second layer weights are all positive, i.e., $v_k > 0$ for all $k \in [l]$. Then,*

    *1. the classification region $R := \{\boldsymbol{x} \in \mathbb{R}^d \mid \mathcal{N}(\boldsymbol{x}) < 0\}$ is a convex polytope. Specifically, the subset $S := \{\boldsymbol{x} \in \mathbb{R}^d \mid \mathcal{N}(\boldsymbol{x}) = v_0\}$ is a convex polytope with $l$ faces.*

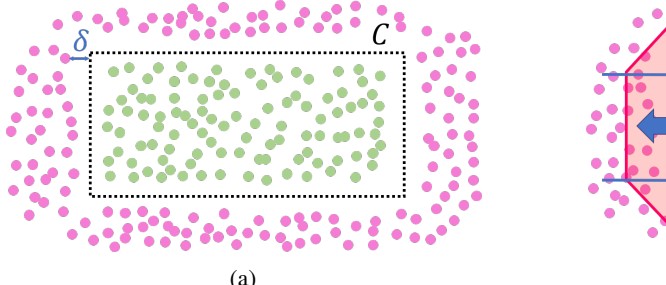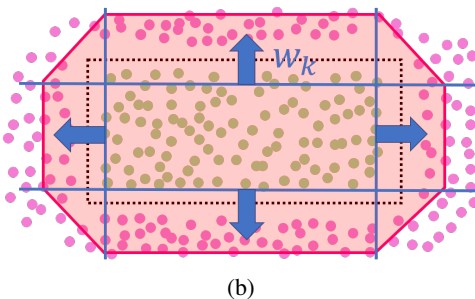

|  (a)  |  (b)  |

Figure 3: Assumptions for the dataset and the network initialization. (a) Dataset $\mathcal{D}$ and a convex polytope $C$ satisfy the Assumption 4.2. (b) One example of network initialization satisfying Assumption 4.2. The red line displays the decision boundary ($\{\boldsymbol{x} \mid \mathcal{N}(\boldsymbol{x}) = 0\}$).

*2. Let $\mathcal{T}$ be a three-layer ReLU network defined by*

$$\mathcal{T}(\boldsymbol{x}) := \min\{\ \mathcal{N}_1(\boldsymbol{x}), \cdots, \mathcal{N}_m(\boldsymbol{x})\ \}. \tag{5}$$

*where each $\mathcal{N}_i$ is a two-layer network defined above, therefore, has positive second layer weights. Then, the classification region $\{\boldsymbol{x} \in \mathbb{R}^d \mid \mathcal{T}(\boldsymbol{x}) < 0\}$ is a union of polytopes.*

The proof and the numerical results with real data with more detailed explanation can be found in Section E in Appendix. The results demonstrate that each class in MNIST, Fashion-MNIST, and CIFAR10 can be separated by at most three convex polytopes with only a few faces (Table 1).

## 4 CONVERGENCE ON THE PROPOSED NETWORKS

In this section, we investigate whether gradient descent can converge to the networks that are induced by the theory in the previous section. Specifically, we focus on the two-layer ReLU networks which are the basic building blocks of the constructions. Let $\mathcal{N}$ be a two-layer ReLU neural network defined in (1), where $\Theta := \{v_0\} \cup \{v_k, \boldsymbol{w}_k, b_k\}_{k \in [l]}$ denotes the set of parameters of $\mathcal{N}$. The mean squared error (MSE) loss and binary cross entropy (BCE) loss are defined by

$$L_{MSE}(\Theta) := \frac{1}{2n} \sum_{i=1}^{n} \Big( \mathcal{N}(\boldsymbol{x}_i) - y_i \Big)^2, \tag{6}$$

$$L_{BCE}(\Theta) := -\frac{1}{n} \sum_{i=1}^{n} \Big( y_i\ \texttt{SIG} \circ \mathcal{N}(\boldsymbol{x}_i) + (1 - y_i)(1 - \texttt{SIG} \circ \mathcal{N}(\boldsymbol{x}_i)) \Big). \tag{7}$$

We now employ a notion of 'polyhedrally separable' dataset from learning theory (Astorino & Gaudioso, 2002; Manwani & Sastry, 2010; Kantchelian et al., 2014), which is an extension of 'linearly separability' and refers to a dataset whose classes can be separated by a convex polytope (Figure 3).

**Definition 4.1.** *Let $\mathcal{D} = \{(\boldsymbol{x}_i, y_i)\}_{i=1}^{n}$ be the given dataset where $\boldsymbol{x}_i \in \mathbb{R}^d$ and $y_i \in \{0, 1\}$. We call that the dataset $\mathcal{D}$ is* polyhedrally separable by $C$ *if there exists a convex polytope $C$ such that $\boldsymbol{x}_i \in C$ if and only if $y_i = 1$ for all $i \in [n]$.*

We further introduce two notations. First, for a convex polytope $C$ composed of $l$ faces, we denote its $k$-th face by $\partial C_k$. Similarly, $\partial^2 C_k$ denotes the boundary of $\partial C_k$, which refers to the 'edge' part of $C$. Second, for a set $A \subset \mathbb{R}^d$, we denote $\#(A) := |\{\boldsymbol{x}_i \mid \boldsymbol{x}_i \in A\}|$ as the number of data points $\boldsymbol{x}_i \in \mathcal{D}$ in the set $A$. We further need the following assumption, on the dataset $\mathcal{D}$ and network initialization.

**Assumption 4.2** (Dataset and initialization assumption)**.** *Suppose the dataset $\mathcal{D}$ is polyhedrally separable by a convex polytope $C$, which consists of $l$ faces and strictly contains the origin point. Let $\delta > 0$ be the minimum distance between $\boldsymbol{x}_i$ and $\partial C$, and $l_k$ be the distance between $\partial C_k$ and the origin point. Then, there exist constants $\rho, R > 0$ such that for any $k \in [l]$ and $\delta < r < R$,*

$$\#\Big(\mathcal{B}_{2r}(\partial^2 C_k)\Big) \le \rho\ \#\Big(\mathcal{B}_{r-\delta}(\partial C_k)\Big). \tag{8}$$

*Furthermore, the parameters $\{(\boldsymbol{w}_k, b_k, v_k)\}_{k \in [l]}$ of a two-layer ReLU network $\mathcal{N}$ defined in (1) are initialized such that $\boldsymbol{w}_k$ are normal to $\partial C_k$ with outward direction, and satisfying*

$$l_k - R \; < \; l_k + \frac{v_0}{v_k \, \|\boldsymbol{w}_k\|} \; < \; -\frac{b_k}{\|\boldsymbol{w}_k\|} \; < \; l_k. \tag{9}$$

The dataset assumption (8) suggests that the data points in the set $\mathcal{B}_r(\partial C)$ for small $r$ are predominantly located in close proximity to the faces of the polytope $C$, rather than its corners (Figure 3(a)). The network initialization assumption implies that every neuron $(\boldsymbol{w}_k, b_k)$ of $\mathcal{N}$ is initialized near $\partial C_k$ as described in Figure 3(b). With these assumptions, we can provide a discrete path that strictly decrease the loss value to zero.

**Theorem 4.3.** *Suppose the dataset $\mathcal{D}$ and the two-layer network (1) satisfy Assumption 4.2. Then,*

1. *for the MSE loss defined in (6), suppose $v_0$ is initialized such that*

$$\frac{\rho}{1-\rho} \frac{4l\rho R^2}{\delta^2} < v_0 < 1. \tag{10}$$

*Then, with step size $\eta < \min\left\{\frac{2}{\delta}, \; \frac{2}{lR}, \; \frac{4\rho l}{(1-\rho)R}\right\}$, there exists a discrete path that the loss value (6) strictly decreases to zero.*

2. *For the BCE loss defined in (7), suppose $v_0$ is initialized such that*

$$0 < v_0 < \log\left(\frac{(1-\rho)\delta}{4\rho R} - 1\right). \tag{11}$$

*Then, with step size $\eta < \min\left\{1, \; \frac{4\rho R}{(1-\rho)\delta^2}\right\}$, there exists a discrete path that the loss value (7) strictly decreases to zero.*

The proof of this theorem can be found in Appendix B.5. Theorem 4.3 asserts that both for MSE loss and BCE loss functions, the loss landscape has no local minima in this initialization region. In other words, stochastic (noisy) gradient descent is believed to converge to the global minimum, which has zero error on the dataset $\mathcal{D}$. This result might be understood by identifying the data distribution condition and initialization condition (Assumption 4.2) such that gradient method can converge. In the next section, we empirically verify that gradient descent method indeed converges to the networks we have proposed (Figure 4).

## 5 EXPERIMENTS

We consider two illustrative topological spaces $\mathcal{X}_1$ and $\mathcal{X}_2$ depicted in Figure 4(a) and (d). $\mathcal{X}_1$ can be understood as a simplicial 2-complex in $\mathbb{R}^2$ comprised of two triangles. The second space $\mathcal{X}_2$ is a hexagon with a pentagonal hole, which has a simple polyope-basis cover itself. We undertake both MSE loss (6) and BCE loss (7) functions. For the BCE loss, we follow the architecture with sigmoid activation proposed in Corollary A.2, to ensure trainability.

For the first dataset $\mathcal{X}_1$, Theorem 3.5 and Corollary A.2 suggest that $2 \xrightarrow{\sigma} 6 \xrightarrow{\sigma} 2 \xrightarrow{\texttt{MAX}} 1$ and $2 \xrightarrow{\sigma} 6 \xrightarrow{\sigma} 2 \xrightarrow{\texttt{SIG}} 1$ are feasible architectures on $\mathcal{X}_1$. For a clearer visualization of weight vectors in each layer, we plot the lines of vanishing points for each layer in blue (the 1st hidden layer), red (the 2nd hidden layer), and the grayscale filling color denotes the output range of the trained network. Moreover, the weight vectors in the first layer encapsulate the two triangles, reflecting the geometrical shape of $\mathcal{X}_1$. Similarly, for the second dataset $\mathcal{X}_2$, Theorem 3.4 suggests that $2 \xrightarrow{\sigma} 11 \xrightarrow{\sigma} 2 \xrightarrow{\sigma} 2 \xrightarrow{\sigma} 1$ and $2 \xrightarrow{\sigma} 11 \xrightarrow{\sigma} 2 \xrightarrow{\texttt{SIG}} 1$ are feasible architectures. More specifically, the eleven neurons in the first layer align with the eleven hyperplanes which are boundaries of the outer hexagon and the inner pentagon, as two neurons in the second hidden layer correspond to the two polygons.

These experimental results verify Theorem 4.3 that the networks proposed in Section 3 can indeed be reached to the global minima by gradient descent with either MSE or BCE loss, under suitable initialization position. We provide further convergence results with various initialization conditions in Figure 10, which shows that the suitable initialization position is necessary to achieve the global convergence.

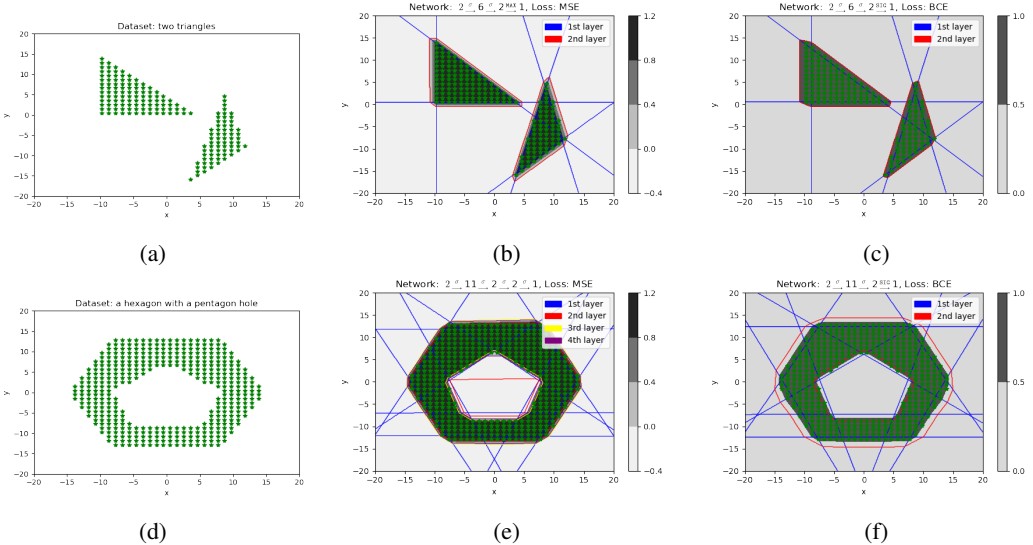

Figure 4: Experimental verification of convergence of gradient descent. (a) and (d) exhibit the shape of two data spaces, which are 'two triangles' and 'a hexagon with a pentagon hole'. (b) and (e) show the converged networks by gradient descent under the MSE loss, where (c) and (f) show the results under the BCE loss. The first layer (blue) and second layer (red) represent the hyperplane and polytopes, respectively. These results verify that gradient descent indeed converges to the networks we proposed in Section 3.

## 6 CONCLUSION

While many previous studies have individually delved into aspects such as the polytope structure of DNNs, the geometric properties of deep ReLU networks, and the bounds on network architecture required for universal approximation, they have often overlooked the interconnections between these properties. In this paper, we have sought to bridge this gap by presenting bounds on network widths that are dependent on the geometric characteristics of the data. Specifically, we established both lower and upper bounds on the widths of shallow networks necessary to approximate the indicator function of a polytope $\mathcal{X}$. Furthermore, we extended these findings to cases where $\mathcal{X}$ has a polytope-basis cover, and obtained a feasible architecture whose widths are determined by the geometric feature of the cover. Similar results were deduced for simplicial $m$-complexes or polytopes with specific shapes of holes, elucidating how the width bound varies according to the complexity of dataset. We also demonstrated both theoretically and empirically that gradient descent can converge to the networks that are induced by our theory, confirming that our theory has practical meanings. Lastly, we applied our findings to investigate geometric shape of real-wordl datasets by training a two-layer network, concluding that they have simple polytope-basis covers.

**Limitations and future work.** One limitation of our work is the assumption of known geometric information about the given datasets, which is often unavailable for real-world datasets. In Section 3.3, detailed in Appendix E, we propose a method to discover a polytope basis-cover for real datasets by training two-layer ReLU networks. However, the optimality of the obtained polytope-basis cover has not been verified, which we leave as a potential avenue for future research.

**Reproducibility.** We note that all our theoretical results are proven in Appendix B. For the detailed experimental setup information, please refer to Appendix D.

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

## A    EXTENSION TO DEEP ReLU NETWORKS AND CROSS ENTROPY LOSS

In Section 3, we presented a variety of three or four-layer neural networks which are feasible architecture on $\mathcal{X}$. In this section, due to its unique structure, we extend our discussion to other neural network architectures. In the following two corollaries, we show the MAX activation or ReLU activation can be substituted by ReLU or SIG, respectively.

Interestingly, the principle outlined in Proposition 3.2 enables us to replace the MAX operation in Theorem 3.5 with an additional ReLU layer. This revelation lays the groundwork for the following corollary, which directly stems from Theorem 3.5

**Corollary A.1** (ReLU networks). *The three-layer neural network in Theorem 3.5 can be changed to a four-layer ReLU network with the architecture* $d \xrightarrow{\sigma} d_1 \xrightarrow{\sigma} k \xrightarrow{\sigma} 1 \rightarrow 1$.

*Proof.* Recall the three-layer neural network proposed in Theorem 3.5, which is $d \xrightarrow{\sigma} d_1 \xrightarrow{\sigma} k \xrightarrow{\text{MAX}} 1$. Let $a_1, \cdots, a_k$ be the input of the last layer, thus the output of the second layer. Since $a_1, \cdots, a_k \in [0, 1]$ from the construction, we get $\text{MAX}(a_1, \cdots, a_k) = 1 - \sigma(1 - a_1 - \cdots - a_k)$. This completes the proof. $\square$

We now turn our attention to neural networks trained under binary cross entropy (BCE) loss defined in (7). For a single pair of data $\boldsymbol{x}$ and its corresponding label $y$, the BCE loss is defined as $\ell(\boldsymbol{x}, y) := y \log(\mathcal{N}(\boldsymbol{x})) + (1-y) \log(1 - \mathcal{N}(\boldsymbol{x}))$. Hence, during training, the output of the neural network must neither be zero nor exceed 1. This requirement is the primary reason classifiers utilize the sigmoid activation function $\text{SIG}(x) := \frac{1}{1+e^{-x}}$.

In light of this, we extend our findings to accommodate a network that employs the sigmoid activation in the final layer. This adjustment can be readily achieved using our previous results, as detailed in Corollary A.1. Notably, this expansion does not necessitate extra layers, unlike in Corollary A.1. The outcome is presented in the subsequent corollary.

**Corollary A.2** (Sigmoid networks). *Let $\mathcal{N}$ be the neural network proposed in one of Proposition 3.2, Theorem 3.4, Theorem 3.5, or Theorem 3.6. Then, the last activation function of $\mathcal{N}$ can be replaced by* SIG.

*Proof.* The proof is similar with the proof of Corollary A.1. If the last layer has MAX activation (Theorem 3.5), then for the inputs $a_1, a_2, \cdots, a_k$, replace $\text{MAX}(a_1, \cdots, a_k)$ to $\text{SIG}(M(-1 + a_1 + a_2 + \cdots + a_k))$ with sufficiently large $M > 0$. If the last layer has ReLU activation (Theorem 3.4, Theorem 3.6, and Proposition 3.2), then just change the last output from $\sigma(a)$ to $Ma$ with sufficiently large $M > 0$. It is easy to verify that these substitutions satisfy the desired property. $\square$

## B    PROOFS

### B.1    PROOF OF PROPOSITION 3.2.

The proof of Proposition 3.2 is divided into two parts. Firstly, we first prove the upper bound by constructing the desired neural network. Secondly, we show the lower bound of widths.

#### B.1.1    THE UPPER BOUND IN PROPOSITION 3.2.

For the given convex polytope $\mathcal{X}$, let $h_1, \cdots, h_l$ be its $l$ hyperplanes enclosing $C$. Let $\boldsymbol{w}_i$ be the unit normal vector of the $i$-th hyperplane $h_i$ oriented inside $C$ (Figure 2(a)). Then the equation of the $i$-th hyperplane $h_i$ is given by $h_i : \{\boldsymbol{x} \mid \boldsymbol{w}_i^\top \boldsymbol{x} + b_i = 0\}$ for some $b_i \in \mathbb{R}$. Let $A_i$ be the intersection of the hyperplane $h_i$ and $C$, which is a face of the polytope $C$. Let $\boldsymbol{x}$ be any point strictly contained in $C$. Since $\boldsymbol{w}_i$ is a unit normal vector, $\boldsymbol{w}_i^\top \boldsymbol{x} + b_i$ refers the distance between the hyperplane $h_i$ and the point $\boldsymbol{x}$. Therefore, the $d$-dimensional Lebesgue measure of $C$ is computed by

$$\mu_d(C) = \frac{1}{d} \sum_{i=1}^{l} (\boldsymbol{w}_i^\top \boldsymbol{x} + b_i) \cdot \mu_{d-1}(A_i) \tag{12}$$

where $\mu_{d-1}$ and $\mu_d$ refer the $(d-1)$ and $d$-dimensional Lebesgue measures, respectively. Note that (12) comes from the volume formula of a convex polytope, which states that the volume is the sum of volume of $l$ pyramids. Then LHS of (12) is constant, which does not depend on the choice of $\boldsymbol{x} \in \mathbb{R}^d$. Now, we define a two-layer ReLU network $\mathcal{T}$ with the architecture $d \xrightarrow{\sigma} l \to 1$ by

$$\mathcal{T}(\boldsymbol{x}) := 1 + M\left(\mu_d(C) - \sum_{i=1}^{l} \frac{1}{d}\mu_{d-1}(A_i) \cdot \sigma(\boldsymbol{w}_i^\top \boldsymbol{x} + b_i)\right) \tag{13}$$

where $M > 0$ is a constant would be determined later. Note that we have $\mathcal{T}(\boldsymbol{x}) = 1$ for $\boldsymbol{x} \in C$ from the construction. However, considering the negative sign, it is worth noting that the equation (12) also holds for $\boldsymbol{x} \notin C$. In particular, for $\boldsymbol{x} \notin C$, (13) deduces

$$\mathcal{T}(\boldsymbol{x}) = 1 + M\left(\mu_d(C) - \sum_{i=1}^{l} \frac{1}{d}\mu_{d-1}(A_i) \cdot \sigma(\boldsymbol{w}_i^\top \boldsymbol{x} + b_i)\right)$$

$$= 1 + M\left(\mu_d(C) - \sum_{i=1}^{l} \frac{1}{d}\mu_{d-1}(A_i) \cdot (\boldsymbol{w}_i^\top \boldsymbol{x} + b_i) + \sum_{\{i\,:\,\boldsymbol{w}_i^\top \boldsymbol{x} + b_i < 0\}} \frac{1}{d}\mu_{d-1}(A_i) \cdot (\boldsymbol{w}_i^\top \boldsymbol{x} + b_i)\right)$$

$$= 1 + M \sum_{\{i\,:\,\boldsymbol{w}_i^\top \boldsymbol{x} + b_i < 0\}} \frac{1}{d}\mu_{d-1}(A_i) \cdot (\boldsymbol{w}_i^\top \boldsymbol{x} + b_i)$$

$$< 1.$$

Therefore, we conclude that

$$\mathcal{T}(\boldsymbol{x}) = 1 \qquad \text{if } \boldsymbol{x} \in C,$$
$$\mathcal{T}(\boldsymbol{x}) < 1 \qquad \text{otherwise.}$$

Lastly, we determine the constant $M$ in $\mathcal{T}$ to satisfy the remained property. For the given $\varepsilon > 0$, consider the closure of complement of the $\frac{\varepsilon}{2}$-neighborhood of $C$; $D := \overline{\left(\mathcal{B}_{\varepsilon/2}(C)\right)^c}$. Then the previsous result shows that

$$\frac{1}{M}(\mathcal{T}(\boldsymbol{x}) - 1) = \mu_d(C) - \sum_{i=1}^{l} \frac{1}{d}\mu_{d-1}(A_i) \cdot \sigma(\boldsymbol{w}_i^\top \boldsymbol{x} + b_i) \tag{14}$$

is bounded above by 0. Furthermore, (14) is continuous piecewise linear, and has the maximum 0 if and only if $\boldsymbol{x} \in C$. Since $D$ is closed and (14) is strictly bounded above by 0 on $D$, (14) has the finite maximum $m < 0$ on $D$.

$$\frac{1}{M}(\mathcal{T}(\boldsymbol{x}) - 1) \le m < 0 \qquad \text{for } \boldsymbol{x} \in D.$$

Now, choose $M$ to satisfy $M > -\frac{1}{m}$. Then if $\boldsymbol{x} \notin B_\varepsilon(C)$, we have $\boldsymbol{x} \in D$, thus

$$\mathcal{T}(\boldsymbol{x}) = 1 + M\left(\mu_d(C) - \sum_{i=1}^{l} \frac{1}{d}\mu_{d-1}(A_i) \cdot \sigma(\boldsymbol{w}_i^\top \boldsymbol{x} + b_i)\right)$$

$$\le 1 + M \cdot m$$

$$< 0.$$

Therefore, we have constructed a two-layer ReLU network $\mathcal{T}$ with the structure $d \xrightarrow{\sigma} l \to 1$ such that

$$\mathcal{T}(\boldsymbol{x}) = 1 \qquad \text{if } \boldsymbol{x} \in C,$$
$$\mathcal{T}(\boldsymbol{x}) < 1 \qquad \text{if } \boldsymbol{x} \in C^c,$$
$$\mathcal{T}(\boldsymbol{x}) < 0 \qquad \text{if } \boldsymbol{x} \notin B_\varepsilon(C).$$

This completes the proof on the upper bound. Lastly, the minimality of depth is proved in Proposition C.2, which shows that a network with architecture $d \xrightarrow{\sigma} d_1 \to 1$ cannot be a feasible architecture on $\mathcal{X}$ for any value $d_1$. $\qquad\square$

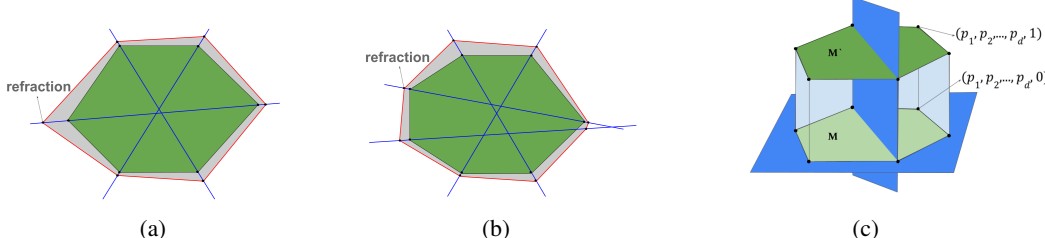

Figure 5: Proof of Proposition 3.2. (a) Given a hexagon which is approximated by 3 hyperplanes given with blue lines and 1 polytope in the second layer given with red hexagon. (b) Given a heptagon which is approximated by 4 hyperplanes given with blue lines and 1 polytope in the second layer given with red heptagon. (c) Hexagon has been extended to the 3-dimensional polytope by incrementing the number of faces by 2 with some potential first layer neurons (blue hyperplanes).

### B.1.2 THE LOWER BOUND IN PROPOSITION 3.2.

Before proving the lower bound, we introduce a definition on *refraction points*. Let $\mathcal{N}(\boldsymbol{x}) := \sigma(v_0 + \sum_{i=1}^{d_1} v_i \sigma(\boldsymbol{w}_i^\top \boldsymbol{x} + b_i))$ be a two-layer network with architecture $d \xrightarrow{\sigma} d_1 \xrightarrow{\sigma} 1$. Then the set of refraction point is defined by

$$\{\boldsymbol{x} \in \mathbb{R}^d \mid \mathcal{N}(\boldsymbol{x}) = 0 \quad \text{and} \quad \boldsymbol{w}_i^\top \boldsymbol{x} + b_i = 0 \quad \text{for some } i \in [l].\}$$

In other words, it is the point where the boundary of a linear partition is 'refracted'.

**Lower bound for $d = 2$ using only refraction points for $k = 1$.** Assume that we are given a convex $l$-gon to approximate. Considering the fact that we can approximate the neural network arbitrarily close, we can see that the approximated second layer, i.e. neural network should have at least $l$ refraction points in order to get the shape of polygon. However, if we look from the perspective of first layer neurons, each line has at most 2 intersection with $l$-gon and it implies that each first layer neuron (or line) can contribute at most 2 meaningful refraction points for the next layer. If we combine above two results, we can obtain that there should be at least $\lceil \frac{l}{2} \rceil$ number of neurons in the first layer, in other words $d_1 \geq \lceil \frac{l}{2} \rceil$. For example, Figure 5(a) and (b) demonstrates the refraction points along with potential first layer hyperplanes (blue lines) and converged polytope at the second layer (red $n$-gon) for hexagon and heptagon, respectively.

**Proof of the optimality when $d = 2$ for $k = 1$.** First of all, we should note that on $\mathbb{R}^2$, any two convex $l$-gon's can be approximated by the same neural network (same $d_1$ value) considering the fact that we can find an approximator for each given error value $\epsilon$. It implies that we can take optimal possible number of neurons in the first layer, which we will denote by $f(l)$ for any given convex $l$-gon. Let's prove that $f(l) = \lceil \frac{l}{2} \rceil$ for $l \geq 5$ along with $f(3) = f(4) = 3$. The cases $l = 3, 4$ should be handled separately, because we trivially need at least $d + 1 = 3$ hyperplanes for any shape (Lemma C.3), so we start the base case from $l \geq 5$ for $d = 2$.

According to the Lemma C.3, it is appearent that for any $l$, the inequality $f(l) \geq 3$ should hold trivially. But if we consider the Figure 5(a), we can observe that one can approximate any hexagon with 3 hyperplanes. Apparently, for any pentagon, quadrilateral, and triangle, we can find a corresponding hexagon to include it as a subfigure and rest of the additional vertices of this hexagon can be shrinked to be almost non-exist. It implies that, same number of hyperplanes approximating hexagon can also approximate the polygons with $l \leq 5$. This final result yields that $f(l) \leq 3$ for $l \leq 6$. If we combine these two findings we can get a nice optimality at the fundamental cases, in other words $f(l) = 3$ for $l \in \{3, 4, 5, 6\}$.

Now, assume the contrary that $f(l) \leq \lceil \frac{l}{2} \rceil - 1$, then it is apparent that there is at least one neuron which contributes to the refraction point of at least 2 vertices (i.e. exactly 2 vertices considering previous discussion). Now, if we remove the chosen neuron and the associated 2 vertices and their edges, then the resulting $(l - 2)$-gon will be approximated by $f(l) - 1$ number of neurons, which implies that $f(l) - 1 \geq f(l - 2)$. Proceeding with the same argument, we can arrive at the conclusion that $f(5)$ or $f(6) \leq 2$; however, we have already proven that $f(5)$ and $f(6)$ are indeed 3. So, the contradiction at the base case yields the result that $f(l) \geq \lceil \frac{l}{2} \rceil$.

For the base cases $n = 5, 6$, we have already demonstrated that $f(5) = f(6) = 3$. Now, take any $l$-gon which has been approximated well with $f(l) = \lceil \frac{l}{2} \rceil$ neurons. Let's add two new vertices to form a new convex polygon with $(l + 2)$ vertices, where the newly added vertices are not adjacent. Then if we add one new neuron which is the line passing through those two points, we can observe that if given $f(l)$ number of neurons approximate $l$-gon, then $f(l) + 1$ can approximate $(l + 2)$-gon by triggering 2 new refraction points. This inductive argument $f(l + 2) \leq f(l) + 1$ yields the result that if we start from $f(5) = f(6) = 3$, we can reach a conclusion that $f(l) \leq \lceil \frac{l}{2} \rceil$. However, we have already shown $f(l) \geq \lceil \frac{l}{2} \rceil$ in the proof above. Therefore, the result follows immediately that the optimal number of neurons in the first hidden layer to approximate any convex polygon with $l$ vertices is $\lceil \frac{l}{2} \rceil$ for $l \geq 5$ and $f(3) = f(4) = 3$. □

**Lower bound for arbitrary dimension $d$ for $k = 1$.** Now we will apply simple induction on the dimensionality to prove the general case for lower bound on the number of first hidden layer neurons. Essentially, we will construct a $d$-dimensional object for $d \geq 2$ such that, one needs at least $d_1 \geq \lceil \frac{l}{2} \rceil + (d - 2)$ number of neurons (hyperplanes) to approximate the convex polytope with $l$ faces. We will proceed with an inductive argument, we have already provided a proof for the base case of $d = 2$ that $d_1 \geq \lceil \frac{l}{2} \rceil$.

*Inductive step.* Suppose that we have a $d$-dimensional convex polytope $M$ with $m$ number of vertices and $l$ number of faces such that the following inequality should hold: $d_1 \geq \lceil \frac{l}{2} \rceil + (d - 2)$. Let's consider the object on $(d+1)$-dimensional space by adding new entry at the end of each coordinate, i.e. any point $(p_1, p_2, ..., p_d)$ on the object will be replaced by the point $(p_1, p_2, ..., p_d, 0)$. Then consider the new shape $M_1$ formed by considering the extension of convex polytope $M$ on $(d+1)$-dimensional space with all the points from $\{p = (p_1, p_2, ..., p_d, x) \mid \forall x = [0, 1] \text{ and } (p_1, p_2, ..., p_d) \in M\}$. Then $M_1$ will lie on $(d + 1)$-dimensional space and it will have $2m$ vertices and $(l + 2)$ number of faces, of which $l$ will be determined by the extensions of faces of polytope $M$ along with two faces from $M$ and its duplicate $M'$. We can also observe the inductive incrementing idea through the Figure 5(c), in which polytope $M$ at $d = 2$ with 6 faces has been extended to the 3-dimensional polytope with 6+2=8 faces.

If we take a closer look at this construction, we can observe that if we take the intersection of each hyperplane from $d_1$ neurons designed for the approximation of $M_1$ and polytope $M$, then those intersections will be hyperplane for $d$-dimensional polytope $M$. It implies that in order to approximate $l$ faces of new polytope, the intersections themselves should approximate the $l$ faces of $M$. Furthermore, other than those $l$ faces formed by faces of previous polytope $M$, we should also consider the other 2 faces, namely $M$ and its duplicate. Those two parallel hyperplanes will require additional 2 neurons to trigger new refraction points for their approximation. Therefore, there should be at least $d_1 \geq \lceil \frac{l}{2} \rceil + (d - 2) + 2$ number of neurons, in which right-hand-side can be equivalently written as $\lceil \frac{l}{2} \rceil + d = \lceil \frac{l+2}{2} \rceil + (d + 1 - 2)$. So, we were able to prove that to have a neural network of the form $d \xrightarrow{\sigma} d_1 \xrightarrow{\sigma} 1$ to approximate the convex polytopes with $l$ faces arbitrarily close, then universally the value of $d_1$ should at least $\lceil \frac{l}{2} \rceil + (d - 2)$.

The result can be also stated that for all $l \geq 2d + 1$ one can find a $d$-dimensional convex polytope with $l$ faces such that the minimum required number neurons in the first hidden layer is at least $\lceil \frac{l}{2} \rceil + (d - 2)$. For $l = 2d - 1$ and $l = 2d$, the lower bound becomes $d_1 \geq 2d - 1$ as we have already described that $f(3) = f(4) = 3$. The lower bound on $l$ comes from the fact that the construction has an inductive fashion to create a new object from previous one by adding 2 new faces in each step. For the rest of the values of number of faces $l$, i.e. $l < 2d - 1$, one can consider the trivial bound of $d + 1$. More strongly, in case of 2-dimensional space, the statement has been proven for all convex polygons that optimal value is indeed $d_1 = \lceil \frac{l}{2} \rceil$.

**Generalization to arbitrary dimension $d$ and depth $k$.** In the context of manifold representations shaped as convex polytopes with varying depths, we employ an inductive approach to establish lower bounds. Leveraging prior findings on two-layer neural networks, we derive insights applicable to arbitrary dimensions $d$. For any given hyperplane in this setting, a maximum of two distinct refraction points can be identified, a premise that underpins our assumption that each second-layer neuron constitutes a polytope comprised of faces, with no more than twice the number of hyperplanes as the first layer. This result has also been used in the proof of Theorem 3.6 and we can observe the trend from the Figure 7(c).

We transform the general case by considering the facets of second or higher-layer neurons as first-layer neurons (hyperplanes), which represent potential refraction points. This transformation allows us to reduce the problem to a two-layer network by decreasing the depth while augmenting the number of hyperplanes in the first layer. More precisely, for a given feasible architecture of the form $d \xrightarrow{\sigma} d_1 \xrightarrow{\sigma} d_2 \xrightarrow{\sigma} \cdots \xrightarrow{\sigma} d_k \xrightarrow{\sigma} 1$, each of $d_2$ number of second layer neurons can contribute at most $2d_1$ hyperplanes along with the $d_1$ hyperplanes in the first layer, which implies total of $d_1 + 2d_1d_2 = d_1(2d_2 + 1)$ hyperplanes. In other words, we can transform the above network to another network $d \xrightarrow{\sigma} d_1(2d_2 + 1) \xrightarrow{\sigma} d_3 \xrightarrow{\sigma} \cdots \xrightarrow{\sigma} d_k \xrightarrow{\sigma} 1$ by reducing the depth by 1. By applying the similar process as above, we assert that initial architecture can be effectively transformed into a more robust architecture, $d \xrightarrow{\sigma} d_1(2d_2 + 1)(2d_3 + 1) \ldots (2d_k + 1) \xrightarrow{\sigma} 1$.

Consequently, we can generalize lower bounds for convex polytope representations of varying depths, drawing on the insights gained from our two-layer formulation. The ultimate result yields a powerful lower bound as

$$d_1 \cdot \prod_{j=2}^{k}(2d_j + 1) \geq \begin{cases} \left\lceil \frac{l}{2} \right\rceil + (d - 2), & \text{if} \quad l \geq 2d + 1, \\ 2d - 1, & \text{if} \quad l = 2d - 1, 2d, \\ d + 1, & \text{if} \quad l < 2d - 1. \end{cases}$$

Moreover, the above result is particularly optimal for the case of convex polygons in two dimensions, where $d = 2$ and $k = 1$, as previously discussed. $\qquad\square$

## B.2 PROOF OF THEOREM 3.4

By Proposition 3.2, for each set $A \in \mathcal{C} = \{P_1, \cdots, P_{n_P}, Q_1, \cdots, Q_{n_Q}\}$, we can construct a two-layer ReLU network $\mathcal{T}_A$ with the architecture $d \xrightarrow{\sigma} l_A \xrightarrow{\sigma} 1$ such that $\mathcal{T}_A(\boldsymbol{x}) = 1$ for $\boldsymbol{x} \in A$ and $\mathcal{T}_A(\boldsymbol{x}) = 0$ for $\boldsymbol{x} \notin B_\varepsilon(A)$, where $l_A$ denotes the number of faces of $A$. Let $a_i := \mathcal{T}_{P_i}$ for $i \in [n_P]$ and $b_j := \mathcal{T}_{Q_j}$ for $j \in [n_Q]$. Define the two neurons in the third hidden layer by

$$a := \sigma(1 - a_1 - \cdots - a_{n_P}) \qquad \text{and} \qquad b := \sigma(1 - b_1 - \cdots - b_{n_Q}).$$

Then, defining the last layer by $\sigma(b - a)$, we obtain the desired network $\mathcal{N}$ which has the architecture $d \xrightarrow{\sigma} l \xrightarrow{\sigma} (n_P + n_Q) \xrightarrow{\sigma} 2 \xrightarrow{\sigma} 1$. $\qquad\square$

## B.3 PROOF OF THEOREM 3.5

Let $X_1, X_2, \cdots, X_k$ be the $k$ facets of $\mathcal{X}$. For each facet $X_i$, we can construct a two-layer ReLU network $\mathcal{T}_i$ such that $\mathcal{T}_i(\boldsymbol{x}) = 1$ for $\boldsymbol{x} \in X_i$ and $\mathcal{T}_i(\boldsymbol{x}) < 0$ for $\boldsymbol{x} \notin B_\varepsilon(X_i)$ by Lemma C.3. Then Proposition 3.2 gives a neural network $\mathcal{N}$ with the architecture $d \xrightarrow{\sigma} d_1 \xrightarrow{\sigma} k \xrightarrow{\text{MAX}} 1$ with $d_1 \leq k(d+1)$, such that $\mathcal{N}$ can approximate $\mathbb{1}_{\{\mathcal{X}\}}$ arbitrarily close. The remaining goal is to reduce the width of the first layer.

From the construction, we recall that $d_1 \leq k(d + 1)$ comes from the fact where each simplex $X_i$ is covered by a $d$-simplex which has $(d + 1)$ hyperplanes. Now consider two $m$-simplices in $\mathbb{R}^d$. If $2m + 2 \leq d + 1$, then we can connect all points of the two $m$-simplices in $\mathbb{R}^d$, and it becomes a $(2m + 2)$-simplex $\Delta^{2m+2}$. Now construct a $d$-simplex $\Delta^{d+1}$ by choosing $(d + 1) - (2m + 2)$ points in $\mathcal{B}_\varepsilon(\Delta^{2m+2})$, whose base is this $(2m + 2)$-simplex. Then, by adding two distinguishing hyperplanes at last, we totally consume $(d + 3)$ hyperplanes to separate two $m$-simplices.

Now we apply this argument to each pair of two simplices. The above argument shows that two $m$-simplices separately covered by $2(d + 1)$ hyperplanes can be re-covered by $(d + 3)$ hyperplanes if $m \leq \lfloor \frac{d-1}{2} \rfloor$, which reduces $(d - 1)$ number of hyperplanes. In other words, we can save $(d - 1)$ hyperplanes for each pair of two $m$-simplices whenever $m \leq \lfloor \frac{d-1}{2} \rfloor$. This provides one improved upper bound of $d_1$:

$$d_1 \quad \leq \quad k(d + 1) - (d - 1) \left\lfloor \frac{1}{2} \sum_{j=0}^{\lfloor \frac{d-1}{2} \rfloor} k_j \right\rfloor. \tag{15}$$

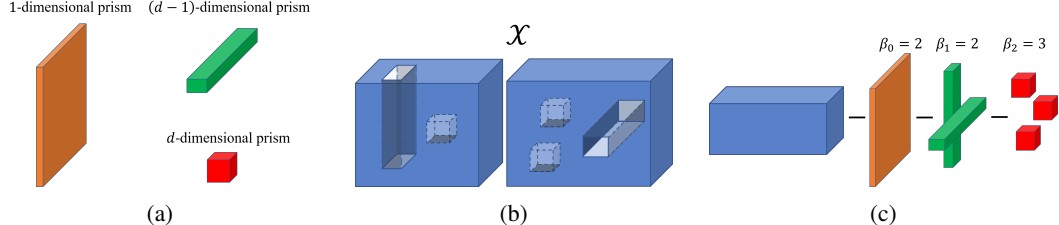

Figure 6: Proof of upper bounds in Theorem 3.6. (a) Some examples of high dimensional prisms. (b) $\mathcal{X}$ is a topological space satisfying the assumption in Theorem 3.6. (b) The removed high dimensional prisms from $\mathcal{X}'$ are displayed. Theorem 3.6 demonstrates that $3 \xrightarrow{\sigma} 34 \xrightarrow{\sigma} 7 \xrightarrow{\sigma} 2 \xrightarrow{\sigma} 1$ is a feasible architecture on $\mathcal{X}$.

Now, we consider another pairing. For $0 \leq j \leq m$, $\mathcal{X}$ has $k_j$ $j$-simplex facets. Since each $j$-simplex has $(j+1)$ points, in particular, a $d$-simplex consists of $(d+1)$-points. Therefore, all points in $\left\lfloor \frac{d+1}{j+1} \right\rfloor$ many $j$-simplices can be contained in one $d$-simplex. In this case, these $j$-simplices can be covered by adding $\left\lfloor \frac{d+1}{j+1} \right\rfloor$ hyperplanes more. Thus if we have $k_j$ many $j$-simplices, then the required number of hyperplanes to separately encapsulate the $j$-simplices is less than or equal to

$$\#\,(\text{the number of } d\text{-simplices}) \cdot \#\,(\text{the required number of hyperplanes in each } d\text{-simplex})$$

$$= \left( \left\lfloor \frac{k_j}{\left\lfloor \frac{d+1}{j+1} \right\rfloor} \right\rfloor + 1 \right) \cdot \left( d+1+ \left\lfloor \frac{d+1}{j+1} \right\rfloor \right)$$

$$\leq \left( k_j \frac{j+1}{d-j} + 1 \right) \cdot \left( d+1+ \frac{d+1}{j+1} \right)$$

$$< (d+1) \left( \frac{j+2}{j+1} \right) \left( k_j \frac{j+1}{d-j} + 1 \right)$$

$$= (d+1) \left( k_j \frac{j+2}{d-j} + \frac{j+2}{j+1} \right) \tag{16}$$

where the inequality is reduced from the property of the floor function: $a - 1 < \lfloor a \rfloor \leq a < \lfloor a \rfloor + 1$ for any $a \in \mathbb{R}$. Then another upper bound of $d_1$ is obtained by applying (16) for all $j \leq m$. However, further note that (16) is greater than the known upper bound $k(d+1)$ if $j > \frac{d}{2}$; the sharing of covering simplex is impossible in this case. Therefore, the upper bound of $d_1$ is given by

$$d_1 \quad \leq (d+1) \sum_{j \leq \frac{d}{2}} \left( k_j \frac{j+2}{d-j} + \frac{j+2}{j+1} \right) + (d+1) \sum_{j > \frac{d}{2}} k_j$$

$$= (d+1) \left[ \sum_{j \leq \frac{d}{2}} \left( k_j \frac{j+2}{d-j} + \frac{j+2}{j+1} \right) + \sum_{j > \frac{d}{2}} k_j \right] \tag{17}$$

To sum up, from (15) and (17), we get the desired result

$$d_1 \leq \min \left\{ k(d+1) - (d-1) \left\lceil \frac{1}{2} \sum_{j=0}^{\left\lfloor \frac{d-1}{2} \right\rfloor} k_j \right\rceil, \; (d+1) \left[ \sum_{j \leq \frac{d}{2}} \left( k_j \frac{j+2}{d-j} + \frac{j+2}{j+1} \right) + \sum_{j > \frac{d}{2}} k_j \right] \right\}.$$

$\square$

## B.4    Proof of Theorem 3.6.

The proof consists of two parts: we prove the upper bound first, and second, we show the lower bound.

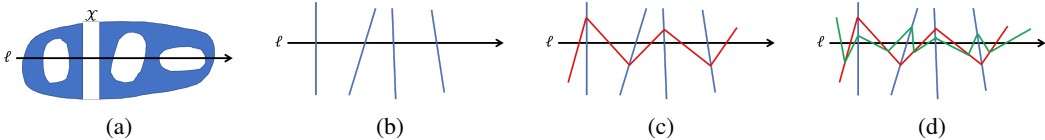

(a)  (b)  (c)  (d)

Figure 7: Proof of lower bounds in Theorem 3.6. (a) Consider a topological space $\mathcal{X}$ whose holes intersect with a straight line $\ell$. (b) $d_1$ neurons in the first hidden layer of $\mathcal{N}$ (blue color) have at most $d_1$ intersection points with $\ell$. (c) A neuron in the second layer (red color) has at most $d_1 + 1$ intersection points with $\ell$. (d) Similarly, a neuron in the third layer (green color) has at most $d_2(d_1 + 1) + 1$ intersection points with $\ell$.

### B.4.1 THE UPPER BOUND IN THEOREM 3.6.

We establish a terminology about the shape of prisms. A prism in $\mathbb{R}^3$ consists of a 'base' and 'height' dimensions, and we generalize it to high dimensional prisms. We define a $k$-**dimensional prism** in $\mathbb{R}^d$ as a topological space homeomorphic to $K \times \mathbb{R}^{d-k}$, where $K \subset \mathbb{R}^k$ is a compact set which is the 'base' of the prism. Roughly speaking, an 1-dimensional prism is a thick 'hyperplane' in $\mathbb{R}^d$, $(d-1)$-dimensional prism is a long 'rod,' and a $d$-dimensional prism is just a hypercube $[0,1]^d$ (Figure 6(a)). Then, for $k = 1, 2, \cdots, d$, removing a $k$-dimensional prism from $\mathcal{X}$ generates a $k$-dimensional hole, which increases $\beta_{k-1}$.

Now, we prove the theorem. Since $\mathcal{X}$ can be described of union and difference of $\sum_{k=0}^{d-1} \beta_k$ convex polytopes, applying Theorem 3.4 provides

$$d \xrightarrow{\sigma} d_1 \xrightarrow{\sigma} \left( \sum_{k=0}^{d-1} \beta_k \right) \xrightarrow{\sigma} 2 \xrightarrow{\sigma} 1$$

is one upper bound of architecture of neural network, where

$$d_1 \leq l + l \cdot \left( \sum_{k=0}^{d-1} \beta_k - 1 \right)$$

$$= l \left( \sum_{k=0}^{d-1} \beta_k \right).$$

For $1 \leq k < d$, $\beta_k$ means the number of $k$-dimensional holes in $\mathcal{X}$, which was made by punching out a $k$-dimensional prism. Since $k$-dimensional prisms have $2k$ faces that penetrate $\mathcal{X}$, we can reduce $2(d - k - 1)$ number of hyperplanes that cover the hole. When $k = 0$, it is easy to check that $2(\beta_0 - 1)$ hyperplanes are required to separate $\beta_0$ connected components. For example, Figure 6(c) shows this process for a topological space given in Figure 6(b). Then, the required total number of hyperplanes is bounded by

$$d_1 \leq l + 2(\beta_0 - 1) + \sum_{k=1}^{d-1} \left( l - 2(d - k - 1) \right) \beta_k$$

which completes the proof. □

### B.4.2 THE LOWER BOUND IN THEOREM 3.6.

Suppose the given architecture $d \xrightarrow{\sigma} d_1 \xrightarrow{\sigma} d_2 \xrightarrow{\sigma} \cdots \xrightarrow{\sigma} d_k \xrightarrow{f} 1$ is a universally feasible architecture on any topological space $\mathcal{X}$ satisfying the assumptions stated in Theorem 3.6. Then, it is enough to consider the 'worst' case of dataset to prove a lower bound. We will use the same idea in the proof of Proposition C.4. Specifically, for the given Betti numbers $\beta_k$, we consider a topological space $\mathcal{X}$ such that every 'hole' intersects with a straight line, say $\ell$. Since each hole intersects with $\ell$ at least two points, we conclude that $\mathcal{N}$ has at least $2\sum_{k=0}^{d} \beta_k$ piecewise linear regions on $\ell$ (Figure 7(a)).

Now we introduce one terminology: from the piecewise linearity of deep ReLU networks, we define a *linear partition region* to be a maximum connected component where the network is affine on. Note

also that the boundary of each linear partition region is non-differentiable points of $\mathcal{N}$ in $\mathbb{R}^d$, which are vanished points of some hidden layers.

We establish the proof by computing the upper bounds of number of linear partition regions on the straight line $\ell$ made by $\mathcal{N}$. For $d_1$ neurons in the first hidden layer, the set of vanishing points are $d_1$ hyperplanes in $\mathbb{R}^d$, thus it can intersect with $\ell$ at most $d_1$ times (Figure 7(b)). Then, consider the vanishing points of the second hidden layer. These points form a bent hyperplane in $\mathbb{R}^d$, which is refracted on the intersection with a vanishing hyperplane of the first layer (Figure 7(c)). Therefore, a vanishing hyperplane of the second hidden layer can intersect with $\ell$ at most $(d_1 + 1)$ times for each neuron. This concludes that the number of vanishing hyperplanes of the second hidden layers can intersect with $\ell$ at most $d_2(d_1 + 1)$ times. By the same arguments, after the third layer, the number of maximum partitions on $\ell$ is bounded by $d_3(d_2(d_1 + 1) + 1) + 1$ (Figure 7(d)), and so on. Then, for the given architecture $d \xrightarrow{\sigma} d_1 \xrightarrow{\sigma} d_2 \xrightarrow{\sigma} \cdots \xrightarrow{\sigma} d_k \xrightarrow{f} 1$, the number of linear partition regions on $\ell$ is bounded by

$$1 + d_k + d_k d_{k-1} + d_k d_{k-1} d_{k-2} + \cdots + d_k \cdots d_1$$
$$= 1 + \sum_{i=1}^{k} \prod_{j=i}^{k} d_j.$$

Therefore, to approximate $\mathbb{1}_{\{\mathcal{X}\}}$ arbitrarily close, we get

$$1 + \sum_{i=1}^{k} \prod_{j=i}^{k} d_j \geq 2 \sum_{k=0}^{d} \beta_k - 1,$$

which completes the proof. $\qquad\square$

## B.5 PROOF OF THEOREM 4.3.

### B.5.1 PROOF FOR MSE LOSS (6).

The proof is divided into several steps. First, for $k \in [l]$, we define the following sets:

$$A_k := \{\boldsymbol{x} \in \mathbb{R}^d \mid \boldsymbol{w}_k^\top \boldsymbol{x} + b_k > 0\} \tag{18}$$
$$B_k := \{\boldsymbol{x} \in \mathbb{R}^d \mid \boldsymbol{w}_k^\top \boldsymbol{x} + b_k > 0 \text{ and } \boldsymbol{w}_j^\top \boldsymbol{x} + b_j > 0 \text{ for } j \neq k\} \tag{19}$$

I.e., $A_k$ is the region where $k$-th neuron is alive, and $B_k$ is the region where only $k$-th neuron is alive (see Figure 8(b,c)). Similarly, we define

$$A_0 := \{\boldsymbol{x} \in \mathbb{R}^d \mid \boldsymbol{w}_k^\top \boldsymbol{x} + b_k < 0\}$$

which is the region where all neurons are dead, except the last bias term $v_0$. Now, we define the following values for every $k \in [l]$:

$$l_k := \text{the distance between } O \text{ and } \partial C_k,$$
$$s_k := -\frac{b_k}{\|\boldsymbol{w}_k\|}, \tag{20}$$
$$t_k := -\frac{v_0}{v_k \|\boldsymbol{w}_k\|}, \tag{21}$$
$$t := \max_{k \in [l]} \{t_k, \delta\}.$$

Then, the network initialization condition (9) gives

$$0 < t_k < R,$$
$$0 < s_k < l_k < s_k + t_k.$$

In other words, $s_k$ is the distance between the origin point $O$ and the hyperplane $\{\boldsymbol{w}_k^\top \boldsymbol{x} + b_k = 0\}$. $t_k$ is the length of 'height' of the region $B_k$ as depicted in Figure 8(c). To be familiar for these notations, we demonstrate the output $\mathcal{N}$ in Figure 8(d) with respect to $\|\boldsymbol{w}_k\|$.

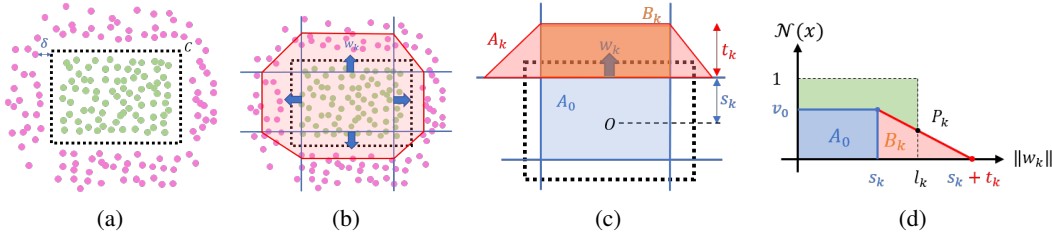

(a)  (b)  (c)  (d)

Figure 8: Proof of Theorem 4.3. (a) The given dataset $\mathcal{D}$ is polyhedrally separable by a black dashed rectangle $C$. (b) Initialization of a two-layer ReLU network $\mathcal{N}$. (c) For $k \in [l]$, sets $A_k$ and $B_k$ defined in (18) and (19) are illustrated. (d) A sideview of the function $\mathcal{N}$ with respect to $\|\boldsymbol{w}_k\|$. $s_k$ and $t_k$ are defined in (20) and (21). Note that the intersection point $P_k$ is invariant after the update of parameters.

It is clear that $\mathcal{N}(\boldsymbol{x}) = v_0$ if $\boldsymbol{x} \in A_0$, and it linearly decreases to zero for $\boldsymbol{x} \in B_k$. When $\boldsymbol{x}_i \in B_k$ satisfies $\boldsymbol{x}_i^\top \frac{\boldsymbol{w}_k}{\|\boldsymbol{w}_k\|} = s_k + t_k$, $\mathcal{N}(\boldsymbol{x}_i) = 0$. Now we are ready to prove the theorem.

For the previously defined sets $A_k$ and $B_k$, the MSE loss (6) is computed by

$$
\begin{aligned}
L_{MSE} &= \frac{1}{2n} \sum_{i=1}^{n} (\mathcal{N}(\boldsymbol{x}_i) - y_i)^2 \\
&= \frac{1}{2n} \sum_{\boldsymbol{x}_i \in A_0} (\mathcal{N}(\boldsymbol{x}_i) - y_i)^2 + \frac{1}{2n} \sum_{\boldsymbol{x}_i \in \cup_k B_k} (\mathcal{N}(\boldsymbol{x}_i) - y_i)^2 + \frac{1}{2n} \sum_{\boldsymbol{x}_i \in \cup_k (A_k \setminus B_k)} (\mathcal{N}(\boldsymbol{x}_i) - y_i)^2 \\
&=: L_1 + L_2 + L_3.
\end{aligned}
\tag{22}
$$

Note that we omitted $\Theta$ notation, the set of all learnable parameters. We will observe the change of these loss values with respect to one update of parameters. We add prime ($'$) for the updated parameter. For the given step size $\eta$, we explicitly provide the update of parameters by

$$
\begin{aligned}
v_0 &\rightarrow v_0' := v_0 + \Delta v_0, \\
s_k &\rightarrow s_k' := s_k + \Delta s_k, \\
t_k &\rightarrow t_k' := t_k + \Delta t_k
\end{aligned}
$$

for all $k \in [l]$, where

$$
\Delta v_0 := \begin{cases} 0 & \text{if} \quad \#(\cup_{k \in [l]} A_k) > 0, \\ -\frac{1}{2}(v_0 - 1)t\eta, & \text{otherwise,} \end{cases}
\tag{23}
$$

$$
\Delta s_k := \begin{cases} \frac{\eta t_k^2}{v_0 + \eta t_k} \cdot \frac{l_k - s_k}{l_k + t_k - s_k} & \text{if} \quad \#(A_k) > 0, \\ 0 & \text{otherwise,} \end{cases}
\tag{24}
$$

$$
\Delta t_k := \begin{cases} \Delta s_k - \frac{\eta t_k^2}{v_0 + \eta t_k} & \text{if} \quad \#(A_k) > 0, \\ \frac{t_k \Delta v_0 - \eta t_k^2}{v_0 + \eta t_k} & \text{otherwise.} \end{cases}
\tag{25}
$$

Specifically, $v_0$ is updated if and only if $\cup_{k \in [l]} A_k$ contains a data point, where $s_k$ and $t_k$ are updated exclusively. The given update terms are proposed to have some invariant quantity. In Figure 8(d), we set $P_k$ to be the output value of $\mathcal{N}$ at $l_k$, and update equations in (23)$\sim$(25) are determined to keep this value $P_k$. Furthermore, it satisfies that the change of the slope is exactly $-\eta$, i.e.,

$$
\begin{aligned}
\Delta\left(-\frac{v_0}{t_k}\right) &:= -\frac{v_0 + \Delta v_0}{t_k + \Delta t_k - \Delta s_k} + \frac{v_0}{t_k} \\
&= -\eta.
\end{aligned}
$$

Note also that $v_0 < 1$ and $s_k < l_k$ are increasing, where $t_k > 0$ is decreasing.

In the subsequent steps, we examine the change of each loss value. The main idea of the proof is computing lower bounds on the reduction of the loss value resulting from one-step update given by (23)$\sim$(25). It is divided into four steps.

**STEP 1.** First, we consider when $\#(\cup_{k \in [l]} A_k) = 0$. In this case, since $L_2 = L_3 = 0$ from the definition (22), it is enough to investigate the change of $L_1$. Recall that

$$L_1 := \frac{1}{2n} \#(A_0)(v_0 - 1)^2.$$

By one-step update of parameters, it becomes $L_1 \rightarrow L_1' := L_1 + \Delta L_1$. Then,

$$
\begin{aligned}
\Delta L_1 &= L_1' - L_1 \\
&= \frac{1}{2n} \sum_{\boldsymbol{x}_i \in A_0} (v_0 + \Delta v_0 - 1)^2 - \frac{1}{2n} \sum_{\boldsymbol{x}_i \in A_0} (v_0 - 1)^2 \\
&= \frac{1}{2n} \cdot (2v_0 - 2 + \Delta v_0) \Delta v_0 \cdot \#(A_0) \\
&= -\frac{\#(A_0)}{n} (1 - v_0) \Delta v_0 + \frac{\#(A_0)}{2n} (\Delta v_0)^2 \\
&= -\frac{\#(A_0)}{2n} (1 - v_0)^2 (t\eta - \frac{1}{4} t^2 \eta^2) \\
&< -\frac{\#(A_0)}{2n} (1 - v_0)^2 \cdot \frac{1}{2} t\eta.
\end{aligned}
$$

Note that we use $\eta < \frac{2}{t} < \frac{2}{\delta}$ on the last inequality. Then, we get

$$
\begin{aligned}
L_1' &= L_1 + \Delta L_1 \\
&= \left(1 + \frac{\Delta L_1}{L_1}\right) L_1 \\
&< \left(1 - \frac{\frac{1}{2n} \#(A_0)(1 - v_0)^2 \cdot \frac{1}{2} t\eta}{\frac{1}{2n} \#(A_0)(1 - v_0)^2}\right) L_1 \\
&= \left(1 - \frac{1}{2} t\eta\right) L_1 \\
&\le \left(1 - \frac{1}{2} \delta\eta\right) L_1
\end{aligned}
\tag{26}
$$

which states that $L_1$ strictly decreases after the update.

**STEP 2.** Now, we consider when $\#(\cup_{k \in [l]} A_k) > 0$. We investigate the second term in (22), defined by

$$L_2 := \frac{1}{2n} \sum_{k \in [l]} \sum_{\boldsymbol{x}_i \in B_k} (\mathcal{N}(\boldsymbol{x}_i) - y_i)^2.$$

Recall that the update of parameters given in (23)$\sim$(25) are chosen to keep $P_k$ value and increasing the absolute value of the slope $-\frac{v_0}{t_k}$ by $\eta$. Therefore, for any $\boldsymbol{x}_i \in B_k$, it $\mathcal{N}(\boldsymbol{x}_i)$ increases (or decreases) if and only if $\boldsymbol{x}_i \in B_k \cap C$ (or $\boldsymbol{x}_i \in B_k \backslash C$). Therefore, $|\mathcal{N}(\boldsymbol{x}_i) - y_i|$ always strictly decreases after the update, which implies that

$$\Delta L_2 := L_2' - L_2 < 0. \tag{27}$$

**STEP 3.** We observe the last term in (22) when $\#(\cup_{k \in [l]} A_k) > 0$, which is the most technical part in this proof. Recall that

$$L_3 := \frac{1}{2n} \sum_{\boldsymbol{x}_i \in \cup_k (A_k \backslash B_k)} (\mathcal{N}(\boldsymbol{x}_i) - y_i)^2.$$

The goal of this step is showing that the absolute change of $L_3$ is less than it of $L_2$, i.e., $|\Delta L_3| < \Delta L_2$. The idea is based on the sparsity of the data distribution in $\mathcal{B}_r(\partial^2 C)$; near the neighborhood of 'edge' parts of the polytope $C$.

Note that for each $k \in [l]$, obviously we have $(A_k \backslash B_k) \subset \mathcal{B}_t(\partial^2 C_k)$ from the linearity of $\mathcal{N}$ (see Figure 8(c) and (d)). It is also worth noting that if $t_k \le \delta$, then $L_3 = 0$ because there is no $\boldsymbol{x}_i$ in $\cup_{k \in [l]} (A_k \backslash B_k)$, and we have nothing to do. Thus we mostly consider $t_k > \delta$ cases.

Let $\mathcal{N}'$ be the network after the one-step update from $\mathcal{N}$. The difference of output is $\Delta\mathcal{N}(\boldsymbol{x}) := \mathcal{N}'(\boldsymbol{x}) - \mathcal{N}(\boldsymbol{x})$. Recall that parameters $v_0, s_k, t_k$ follow the updated rule (23) $\sim$ (25) such that network have a constant output on $\partial C_k \cap B_k$ (Figure 8(c) and (d)). This implies that both networks $\mathcal{N}$ and $\mathcal{N}'$ have fixed outputs for $\partial C_k \cap B_k$, and then the affine space connecting those fixed points also has the fixed output which comes from the piecewise linearity of $\mathcal{N}$.

**STEP 3-1** First, we compute an upper bound of $|\Delta L_3|$. Before we start, we adopt a terminology 'linear partition' which is widely used in many similar studies (Hanin & Rolnick, 2019). Since $\mathcal{N}(\boldsymbol{x})$ is piecewise linear, we consider the input space partition in $A_k\backslash B_k$ where $\mathcal{N}$ is linear on. Observing the 'corner' parts of the polytope $C$ (see Figure 8(c) and (d)), each partition is intersection of some neurons of $\mathcal{N}$. Choose one partition $P \subset A_k\backslash B_k$, and let $J_P \subset [l]$ be the index set of $P$ that $\boldsymbol{w}_j$ is activated on $P$ if and only if $j \in J_P$, or namely, $P = \bigcap_{j\in J_P} A_j$. Then obviously, the partition $P$ is contained in a ball with radius $\max_{j\in J_P} t_j \le t < R$. On the contrary, any partition $P$ is contained in $t_k$-radius ball from $\partial_k$ for some $k$. Using this, we can disjointly separate the partitions to $\mathcal{Q}_k$ such that

1. $Q_k \subset (A_k\backslash B_k)$
2. Every $P \in \cup_{k\in[l]}(A_k\backslash B_k)$ is exactly contained in one of $Q_k$.
3. Every $P \in Q_k$ can be bounded by a ball with radius $t_k$.

Note that $\mathcal{Q}_k$ is a collection of partitions, which can be empty. Using this, we decompose $L_3$ by the following way. This is just rearranging the terms in $L_3$.

$$L_3 = \frac{1}{2} \sum_{\boldsymbol{x}_i \in \cup_{k\in[l]}(A_k\backslash B_k)} (\mathcal{N}(\boldsymbol{x}_i) - y_i)^2$$

$$= \frac{1}{2} \sum_{k\in[l]} \sum_{\boldsymbol{x}_i \in \mathcal{Q}_k} (\mathcal{N}(\boldsymbol{x}_i) - y_i)^2$$

$$=: \frac{1}{2} \sum_{k\in[l]} L_{3,k}.$$

Now, we bound the change of network output $\Delta\mathcal{N}(\boldsymbol{x}_i)$ for $\boldsymbol{x}_i \in P \in \mathcal{Q}_k$.

$$|\Delta\mathcal{N}(\boldsymbol{x}_i)| = |\mathcal{N}(\boldsymbol{x}_i) - \mathcal{N}'(\boldsymbol{x}_i)|$$

$$\le \left| \sum_{j\in J_P} \Delta\left(-\frac{v_0}{t_j}\right) t_k \right|$$

$$= \sum_{j\in J_P} \eta R$$

$$\le lR\eta.$$

Above inequalities come from the fact that, the change of linear value is bounded by product of the change of slope and the maximum diameter of the set. Finally, for a $k \in [l]$, we compute an upper bound of the loss variation of $L_{3,k}$.

$$|\Delta L_{3,k}| = \left| \frac{1}{2n} \sum_{\boldsymbol{x}_i \in (A_k\backslash B_k)} (\mathcal{N}(\boldsymbol{x}_i) + \Delta\mathcal{N}(\boldsymbol{x}_i) - y_i)^2 - \frac{1}{2n} \sum_{\boldsymbol{x}_i \in (A_k\backslash B_k)} (\mathcal{N}(\boldsymbol{x}_i) - y_i)^2 \right|$$

$$= \left| \frac{1}{n} \sum_{\boldsymbol{x}_i \in (A_k\backslash B_k)} \left(\mathcal{N}(\boldsymbol{x}_i) - y_i + \frac{1}{2}\Delta\mathcal{N}(\boldsymbol{x}_i)\right) \cdot \Delta\mathcal{N}(\boldsymbol{x}_i) \right|$$

$$\le \frac{1}{n} \sum_{\boldsymbol{x}_i \in (A_k\backslash B_k)} \left( |\mathcal{N}(\boldsymbol{x}_i) - y_i| \cdot |\Delta\mathcal{N}(\boldsymbol{x}_i)| + \frac{1}{2}|\Delta\mathcal{N}(\boldsymbol{x}_i)|^2 \right)$$

$$\le \frac{1}{n} \sum_{\boldsymbol{x}_i \in (A_k\backslash B_k)} \left( 1 \cdot |\Delta\mathcal{N}(\boldsymbol{x}_i)| + \frac{1}{2}|\Delta\mathcal{N}(\boldsymbol{x}_i)|^2 \right)$$

$$\leq \frac{1}{n} \# \left( \mathcal{B}_{t_k}(\partial^2 C_k) \right) \cdot (lR\eta + \frac{1}{2} l^2 R^2 \eta^2)$$

$$\leq \frac{2}{n} \# \left( \mathcal{B}_{t_k}(\partial^2 C_k) \right) \cdot lR\eta$$

$$\leq \frac{2lR\eta}{n} \cdot \rho \; \#(\mathcal{B}_{t_k}(\partial C_k)). \tag{28}$$

Note that we used $\eta < \frac{2}{lR}$ to bound the quadratic term $\eta^2$.

**STEP 3-2.** Now, we compute a similar bound for $\Delta L_2$. It can be decomposed to the sum on each $B_k$.

$$L_2 = \frac{1}{2n} \sum_{k \in [l]} \sum_{\boldsymbol{x}_i \in B_k} (\mathcal{N}(\boldsymbol{x}_i) - y_i)^2$$

$$=: \sum_{k \in [l]} L_{2,k}.$$

We use the fact that each data point $\boldsymbol{x}_i$ is far from $\partial C$ at least $\delta$. From the definition, we get $\Delta \mathcal{N}(\boldsymbol{x}_i) > \delta \eta$. Note that if $t_k < 2\delta$, then $L_{3,k}$ strictly decreases and we have nothing to do. Otherwise, when $t_k > 2\delta$, we have $R > 2\delta$ and there is data far from $\delta$ distance from $\partial B_k$. For such data point $\boldsymbol{x}_i$, we get

$$\mathcal{N}(\boldsymbol{x}_i) - 0 = v_0 - \frac{v_0}{t_k}(t_k - \delta)$$

$$= v_0 \frac{\delta}{t_k}$$

$$> \frac{v_0 \delta}{R}$$

and

$$1 - \mathcal{N}(\boldsymbol{x}_i) = 1 - \left( v_0 - \frac{v_0}{t_k}(t_k - \delta) \right)$$

$$= 1 - \frac{v_0}{t_k} \delta$$

$$> 1 - \frac{1}{2} v_0$$

$$> \frac{1}{2} v_0$$

$$> \frac{v_0 \delta}{R}.$$

Therefore, we have shown that

$$\min_{s_k + \delta \leq \|\boldsymbol{x}_i\| \leq s_k + t_k - \delta} |\mathcal{N}(\boldsymbol{x}_i) - y_i| \geq \frac{v_0 \delta}{R}.$$

Now we induce the lower bound of $\Delta L_{2,k}$.

$$\Delta L_{2,k} = \frac{1}{2n} \sum_{\boldsymbol{x}_i \in B_k} \left( (\mathcal{N}(\boldsymbol{x}_i) + \Delta \mathcal{N}(\boldsymbol{x}_i) - y_i)^2 - (\mathcal{N}(\boldsymbol{x}_i) - y_i)^2 \right)$$

$$= \frac{1}{n} \sum_{\boldsymbol{x}_i \in B_k} \left( \mathcal{N}(\boldsymbol{x}_i) - y_i + \frac{1}{2} \Delta \mathcal{N}(\boldsymbol{x}_i) \right) \cdot \Delta \mathcal{N}(\boldsymbol{x}_i)$$

$$= \frac{1}{n} \sum_{\boldsymbol{x}_i \in B_k} \left( -|(\mathcal{N}(\boldsymbol{x}_i) - y_i)| \cdot |\Delta \mathcal{N}(\boldsymbol{x}_i)| + \frac{1}{2} |\Delta \mathcal{N}(\boldsymbol{x}_i)|^2 \right)$$

$$\leq \frac{1}{n} \sum_{\boldsymbol{x}_i \in B_k} \left( - \min_{\delta \leq \|\boldsymbol{x}_i\| - s_k \leq t_k - \delta} |\mathcal{N}(\boldsymbol{x}_i) - y_i| \cdot \eta \delta + \frac{1}{2} R^2 \eta^2 \right)$$

$$\leq -\frac{1}{n} \# \left( \mathcal{B}_{\frac{t_k}{2} - \delta}(\partial C_k) - \mathcal{B}_{t_k}(\partial^2 C_k) \right) \cdot \left( \frac{v_0 \delta}{R} \cdot \delta \eta - \frac{1}{2} R^2 \eta^2 \right)$$

$$\leq -\frac{1}{n}(1-\rho)\#(\mathcal{B}_{t_k}(\partial C_k)) \cdot \left(\frac{v_0 \delta^2 \eta}{R} - \frac{1}{2}R^2\eta^2\right)$$

$$< \frac{1}{n}\frac{(1-\rho)v_0\delta^2\eta}{2R}\,\#(\mathcal{B}_{t_k}(\partial C_k)). \tag{29}$$

Note that Assumption 4.2 on dataset $\mathcal{D}$ is used to induce this inequality. Now, we compare (29) and (28) with initialization condition (10) for $v_0$. Then, we finally get

$$|\Delta L_{3,k}| \leq \frac{2lR\eta\rho}{n}\,\#(\mathcal{B}_{t_k}(\partial C_k))$$

$$< \frac{(1-\rho)v_0\delta^2}{2nR}\eta \cdot \#(\mathcal{B}_{t_k}(\partial C_k))$$

$$< -\Delta L_{2,k}$$

for every $k \in [l]$. By summing up, we conclude $|\Delta L_3| < -\Delta L_2$ or,

$$\Delta L_2 + \Delta L_3 < 0. \tag{30}$$

**STEP 4.** Finally, we combine all results in the previous steps. When $\#(\cup_{k\in[l]} A_k) > 0$, only $L_2$ and $L_3$ are changed, then one step update gives

$$L' = L + \Delta L$$
$$= L_1 + L_2 + L_3 + \Delta L_1 + \Delta L_2 + \Delta L_3$$
$$< L_1 + L_2 + L_3$$

from (27) and (30). Furthermore, since $s_k < l_k$ increases and $t_k > 0$ decreases, the updated parameters satisfy the assumption (9) again. Using mathematical induction, we can repeat above steps until $\#(\cup_{k\in[l]} A_k) = 0$. After achieving $\#(\cup_{k\in[l]} A_k) = 0$, we get $L_2 = L_3 = 0$ from their definition (22). Then, the remained loss $L_1$ exponentially decreases to zero because

$$L' = L_1 + \Delta L_1$$
$$\leq \left(1 - \frac{1}{2}\delta\eta\right)L_1$$
$$\leq \left(1 - \frac{1}{2}\delta\eta\right)L$$

from (26). This completes the proof. $\square$

### B.5.2 Proof for BCE loss (7).

The proof idea is similar with the previous proof. We use the same definitions for $A_k, B_k, s_k, t_k, l_k,$ and other notations. The BCE loss (7) is rearranged by

$$L_{BCE} = -\frac{1}{n}\sum_{i=1}^{n}(y_i \log \mathtt{SIG} \circ \mathcal{N}(\boldsymbol{x}_i) + (1-y_i)\log(1 - \mathtt{SIG} \circ \mathcal{N}(\boldsymbol{x}_i)))$$

$$= -\frac{1}{n}\sum_{\boldsymbol{x}_i \in A_0} \log \mathtt{SIG} \circ \mathcal{N}(\boldsymbol{x}_i)$$

$$- \frac{1}{n}\sum_{k\in[l]}\sum_{\boldsymbol{x}_i \in B_k}(y_i \log \mathtt{SIG} \circ \mathcal{N}(\boldsymbol{x}_i) + (1-y_i)\log(1 - \mathtt{SIG} \circ \mathcal{N}(\boldsymbol{x}_i))) \tag{31}$$

$$- \frac{1}{n}\sum_{k\in[l]}\sum_{\boldsymbol{x}_i \in (A_k\backslash B_k)}(y_i \log \mathtt{SIG} \circ \mathcal{N}(\boldsymbol{x}_i) + (1-y_i)\log(1 - \mathtt{SIG} \circ \mathcal{N}(\boldsymbol{x}_i)))$$

$$=: L_1 + L_2 + L_3.$$

Before we start, we compute the derivative and its bound of some functions. For $\zeta \in \mathbb{R}$, define

$$f(\zeta) := \log \mathtt{SIG}(\zeta),$$

$$g(\zeta) := \log(1 - \texttt{SIG}(\zeta)).$$

Then their derivatives are given by

$$\frac{d}{d\zeta}f(\zeta) := 1 - \texttt{SIG}(\zeta),$$

$$\frac{d}{d\zeta}g(\zeta) := -\texttt{SIG}(\zeta).$$

From the mean value theorem (MVT), we get

$$f(\zeta + \Delta\zeta) = f(\zeta) + f'(\zeta)\Delta\zeta + \frac{1}{2}f''(\tilde{\zeta})(\Delta\zeta)^2$$

$$\geq f(\zeta) + (1 - \texttt{SIG}(\zeta))\Delta\zeta - \frac{1}{2}(\Delta\zeta)^2$$

and

$$f(\zeta + \Delta\zeta) - f(\zeta) = (1 - \texttt{SIG}(\tilde{\zeta}))\Delta\zeta$$
$$\leq \Delta\zeta.$$

Now we defin the update of parameters. For $k \in [l]$, the update of $v_0$ is given by

$$\Delta v_0 := \begin{cases} 0 & \text{if} \quad \#(\cup_{k \in [l]} A_k) > 0, \\ (1 - \texttt{SIG}(v_0))\eta. & \text{otherwise} \end{cases} \tag{32}$$

For $\Delta s_k$ and $\Delta t_k$, we adopt the same update defined in (24) and (25). Namely, the update of parameters preserves the value of $\mathcal{N}$ on $l_k$ and the change of slope is set to $-\eta$. We repeat the analogous arguments in the previous proof.

**STEP 1.** Firstly, we consider the first loss term $L_1$ in (31) when $\#(\cup_{k \in [l]} A_k) = 0$. It is changed by

$$\Delta L_1 = L_1' - L_1$$

$$= -\frac{1}{n}\sum_{\boldsymbol{x}_i \in A_0} \log \texttt{SIG}(v_0 + \Delta v_0) + \frac{1}{n}\sum_{\boldsymbol{x}_i \in A_0} \log \texttt{SIG}(v_0)$$

$$= -\frac{\#(A_0)}{n}(f(v_0 + \Delta v_0) - f(v_0))$$

$$\leq -\frac{\#(A_0)}{n}\left((1 - \texttt{SIG}(v_0))\Delta v_0 - \frac{1}{2}(\Delta v_0)^2\right)$$

$$= -\frac{\#(A_0)}{n}\left((1 - \texttt{SIG}(v_0))^2\eta - \frac{1}{2}(1 - \texttt{SIG}(v_0))^2\eta^2\right)$$

$$< -\frac{\#(A_0)}{n}\frac{1}{2}(1 - \texttt{SIG}(v_0))^2\eta.$$

Therefore, $L_1$ strictly decreases. Note that we used $\eta < 1$ to bound the $\eta^2$ term.

**STEP 2.** Secondly, we consider when $\#(\cup_{k \in [l]} A_k) > 0$. As discussed in the previous subsection, $\mathcal{N}(\boldsymbol{x}_i)$ strictly increases (or decreases) if and only if $y_i = 1$ (or 0, respectively) because the slope $-\frac{v_0}{t_k}$ changes $-\eta$. This shows that $\Delta L_2 < 0$.

**STEP 3.** Thirdly, we observe $\Delta L_2$ and $|\Delta L_3|$ when $\#(\cup_{k \in [l]} A_k) > 0$. We compute a bound of $\Delta L_3$ first. For any $k \in [l]$,

$$|\Delta L_3| = \frac{1}{n}\left| \sum_{\boldsymbol{x}_i \in (A_k \setminus B_k)} y_i(f(\mathcal{N}(\boldsymbol{x}_i) + \Delta\mathcal{N}(\boldsymbol{x}_i)) - f(\mathcal{N}(\boldsymbol{x}_i))) \right.$$

$$\left. + (1 - y_i)(g(\mathcal{N}(\boldsymbol{x}_i) + \Delta\mathcal{N}(\boldsymbol{x}_i)) - g(\mathcal{N}(\boldsymbol{x}_i))) \right|$$

$$\leq \frac{1}{n}\sum_{\boldsymbol{x}_i \in (A_k \setminus B_k)} \left|(f(\mathcal{N}(\boldsymbol{x}_i) + \Delta\mathcal{N}(\boldsymbol{x}_i)) - f(\mathcal{N}(\boldsymbol{x}_i)))\right| + \left|(g(\mathcal{N}(\boldsymbol{x}_i) + \Delta\mathcal{N}(\boldsymbol{x}_i)) - g(\mathcal{N}(\boldsymbol{x}_i)))\right|$$

$$< \frac{1}{n} \sum_{\boldsymbol{x}_i \in (A_k \setminus B_k)} 2|\Delta \mathcal{N}(\boldsymbol{x}_i)|$$

$$< \frac{2}{n} \#(\mathcal{B}_{t_k}(\partial^2 C_k)) \cdot \max_{\boldsymbol{x}_i \in (A_k \setminus B_k)} |\Delta \mathcal{N}(\boldsymbol{x}_i)|$$

$$< \frac{2R\eta}{n} \#(\mathcal{B}_{t_k}(\partial^2 C_k)).$$

We obtain a similar bound for $\Delta L_{2,k}$. Let $V_0 := \log\left(\frac{(1-\rho)\delta}{4\rho R} - 1\right)$ be the upper bound of initialization of $v_0$. Note also that $\mathrm{SIG}(V_0) = 1 - \frac{4\rho R}{(1-\rho)\delta}$ and $\eta < \frac{1-\mathrm{SIG}(v_0)}{\delta}$. Then,

$$\Delta L_{2,k} = -\frac{1}{n} \sum_{\boldsymbol{x}_i \in B_k} \Bigg( y_i(f(\mathcal{N}(\boldsymbol{x}_i) + \Delta \mathcal{N}(\boldsymbol{x}_i)) - f(\mathcal{N}(\boldsymbol{x}_i)))$$

$$+ (1 - y_i)(g(\mathcal{N}(\boldsymbol{x}_i) + \Delta \mathcal{N}(\boldsymbol{x}_i)) - g(\mathcal{N}(\boldsymbol{x}_i)))\Bigg)$$

$$= -\frac{1}{n} \sum_{\boldsymbol{x}_i \in B_k, y_i = 1} \Bigg( (f(\mathcal{N}(\boldsymbol{x}_i) + \Delta \mathcal{N}(\boldsymbol{x}_i)) - f(\mathcal{N}(\boldsymbol{x}_i)))$$

$$- \frac{1}{n} \sum_{\boldsymbol{x}_i \in B_k, y_i = 0} (g(\mathcal{N}(\boldsymbol{x}_i) + \Delta \mathcal{N}(\boldsymbol{x}_i)) - g(\mathcal{N}(\boldsymbol{x}_i)))\Bigg)$$

$$< -\frac{1}{n} \sum_{\boldsymbol{x}_i \in B_k, y_i = 1} \left( (1 - \mathrm{SIG} \circ \mathcal{N}(\boldsymbol{x}_i))\Delta \mathcal{N}(\boldsymbol{x}_i) - \frac{1}{2}(\Delta \mathcal{N}(\boldsymbol{x}_i))^2 \right)$$

$$- \frac{1}{n} \sum_{\boldsymbol{x}_i \in B_k, y_i = 0} \left( -\mathrm{SIG} \circ \mathcal{N}(\boldsymbol{x}_i) \cdot \Delta \mathcal{N}(\boldsymbol{x}_i) - \frac{1}{2}(\Delta \mathcal{N}(\boldsymbol{x}_i))^2 \right)$$

$$< -\frac{1}{n} \sum_{s_k - l_k + \delta < h_i < -\delta} \left( (1 - \mathrm{SIG} \circ \mathcal{N}(\boldsymbol{x}_i))\Delta \mathcal{N}(\boldsymbol{x}_i) - \frac{1}{2}(\Delta \mathcal{N}(\boldsymbol{x}_i))^2 \right)$$

$$- \frac{1}{n} \sum_{\delta < h_i < s_k + t_k - \delta} \left( -\mathrm{SIG} \circ \mathcal{N}(\boldsymbol{x}_i) \cdot \Delta \mathcal{N}(\boldsymbol{x}_i) - \frac{1}{2}(\Delta \mathcal{N}(\boldsymbol{x}_i))^2 \right)$$

$$< -\frac{1}{n} \sum_{s_k - l_k + \delta < h_i < -\delta} \left( (1 - \mathrm{SIG}(V_0))\delta\eta - \frac{1}{2}\delta^2\eta^2 \right)$$

$$- \frac{1}{n} \sum_{\delta < h_i < s_k + t_k - \delta} \left( \mathrm{SIG}(0) \cdot \delta\eta - \frac{1}{2}\delta^2\eta^2 \right)$$

$$< -\frac{1}{n} \# \left( \mathcal{B}_{\frac{t_k}{2} - \delta}(\partial C_k) - \mathcal{B}_{t_k}(\partial^2 C_k) \right) \cdot \left( (1 - \mathrm{SIG}(V_0))\delta\eta - \frac{1}{2}\delta^2\eta^2 \right)$$

$$< -\frac{1}{n}(1 - \rho)\#(\mathcal{B}_{t_k}(\partial C_k)) \cdot \frac{1}{2}(1 - \mathrm{SIG}(V_0))\delta\eta.$$

Therefore,

$$|\Delta L_{3,k}| < \frac{2R\eta}{n}\rho \,\#(\mathcal{B}_{t_k}(\partial C_k))$$

$$< \frac{1}{n}(1 - \rho)\#(\mathcal{B}_{t_k}(\partial C_k)) \cdot \frac{1}{2}(1 - \mathrm{SIG}(V_0))\delta\eta$$

$$< -L_{2,k}$$

and we get $\Delta L_2 + \Delta L_3 < 0$.

**STEP 4.** Finally, we combine results in the previous steps. When $\#(\cup_{k \in [l]} A_k) > 0$, $v_0$ is bounded by $V_0$ and we get $\Delta L_2 + \Delta L_3 < 0$ from **STEP 3**. After update, since $s_k < l_k$ increases and $t_k > 0$ decreases, the updated parameters satisfy Assumption 4.2 again. It is repeated with strictly

decreasing loss until reaching $\#(\cup_{k\in[l]} A_k) = 0$. After that, $v_0$ begins to strictly increase, which strictly decreases all $L_1$, $L_2$, and $L_3$. Further, the update equation 32 provides $v_0$ goes to infinity. Therefore, $\mathcal{N}(\boldsymbol{x}_i) \to \infty$ if and only if it label $y_i = 1$, concludes $L_{BCE}$ converges to zero.

This completes the whole proof of Theorem 4.3. □

## C  ADDITIONAL PROPOSITIONS AND LEMMAS

**Proposition C.1.** *Let $\mathcal{X}$ be a compact set in $\mathbb{R}^d$. For arbitrary $\delta > 0$ and $p > 0$, suppose there exists a function $f_\delta : \mathbb{R}^d \to \mathbb{R}$ such that $f_\delta(\mathbb{R}^d) = [0,1]$ and*

$$f_\delta(\boldsymbol{x}) = 1 \qquad \text{if } \boldsymbol{x} \in \mathcal{X},$$
$$f_\delta(\boldsymbol{x}) = 0 \qquad \text{if } \boldsymbol{x} \notin B_\delta(\mathcal{X}).$$

*Then, for arbitrary $\varepsilon > 0$, there exists a function $\mathcal{N} : \mathbb{R}^d \to \mathbb{R}$ such that*

$$\left\| \mathcal{N}(\boldsymbol{x}) - \mathbb{1}_{\{\mathcal{X}\}}(\boldsymbol{x}) \right\|_{L^p(\mathbb{R}^d)} < \varepsilon. \tag{33}$$

*Proof.* Let $\mu$ be the Lebesgue measure in $\mathbb{R}^d$. First note that

$$\lim_{\delta \to 0^+} \mu(B_\delta(\mathcal{X}) - \mathcal{X}) = \mu(\bar{\mathcal{X}} \backslash \mathcal{X}) = 0.$$

Therefore, for a given $\varepsilon$, there exists $\delta > 0$ such that

$$\mu\left(B_\delta(\mathcal{X}) - \mathcal{X}\right) < \varepsilon^p.$$

From the assumption, for such $\delta$, there exists a function $f_\delta : \mathbb{R}^d \to \mathbb{R}$ that satisfies $f_\delta(\mathbb{R}^d) = [0,1]$, $f_\delta(\boldsymbol{x}) = 1$ for $\boldsymbol{x} \in \mathcal{X}$, and $f_\delta(\boldsymbol{x}) = 0$ if $\boldsymbol{x} \notin B_\delta(\mathcal{X})$. Now, define $\mathcal{N} := f_\delta$. Then,

$$\begin{aligned}
\left\| \mathcal{N}(\boldsymbol{x}) - \mathbb{1}_{\{\mathcal{X}\}}(\boldsymbol{x}) \right\|_{L^p(\mathbb{R}^d)}^p &= \int_{\mathbb{R}^d} |\mathcal{N}(\boldsymbol{x}) - \mathbb{1}_{\{\mathcal{X}\}}(\boldsymbol{x})|^p \, d\mu \\
&= \int_{B_\delta(\mathcal{X})} |\mathcal{N}(\boldsymbol{x}) - \mathbb{1}_{\{\mathcal{X}\}}(\boldsymbol{x})|^p \, d\mu \\
&= \int_{B_\delta(\mathcal{X}) \backslash \mathcal{X}} |\mathcal{N}(\boldsymbol{x}) - \mathbb{1}_{\{\mathcal{X}\}}(\boldsymbol{x})|^p \, d\mu \\
&\leq 1^p \cdot \mu\left(B_\delta(\mathcal{X}) \backslash \mathcal{X}\right) \\
&< \varepsilon^p.
\end{aligned}$$

Therefore, $\mathcal{N}$ is the desired function satisfying (33). □

**Proposition C.2.** *Let $f^* : \mathbb{R}^d \to \mathbb{R}$ be a nonzero compactly supported function. Then for $p \geq 1$ and $d \geq 2$, a network with the architecture $d \xrightarrow{\sigma} d_1 \to 1$ cannot approximate $f^*$ in $L^p(\mathbb{R}^d)$ under some positive error, for any value $d_1$.*

*Proof of Proposition C.2.* We prove by contradiction: suppose two-layer ReLU networks are dense in $L^p(\mathbb{R}^d)$. Then for a nonzero function $f^*$ and $\varepsilon > 0$, there exists a nonzero two-layer ReLU network $f = \sum_{i=1}^{k} v_i \sigma(\boldsymbol{w}_i^\top \boldsymbol{x} + b_i) + b_0$ such that $\|f - f^*\|_{L^p(\mathbb{R}^d)} < \varepsilon$. Without loss of generality, we can assume that all $v_i, \boldsymbol{w}_i$ are nonzero, i.e., $f$ has the minimal representation. Now, let $K$ be a compact (thus bounded) set that contains the support of $f^*$. Since $f$ is piecewise linear and defined on the unbounded domain $\mathbb{R}^d$, we can choose an unbounded partition $A \subset \mathbb{R}^d$ such that $f$ is linear in $A$ and $A \cap K^c$ is unbounded. By re-ordering of indices if needed, there exists $k_A \in \mathbb{N}$ such that $\boldsymbol{w}_i^\top \boldsymbol{x} + b_i \geq 0$ if and only if $i \in [k_A]$. Then for $\boldsymbol{x} \in A$, $f(\boldsymbol{x}) = \sum_{i=1}^{k_A} v_i(\boldsymbol{w}_i^\top \boldsymbol{x} + b_i) + b_0$ and we get

$$\begin{aligned}
\|f - f^*\|_{L^p(\mathbb{R}^d)}^p &= \int_{\mathbb{R}^d} |f - f^*|^p d\mu \\
&\geq \int_{A \cap K^c} |f - f^*|^p d\mu
\end{aligned}$$

$$\geq \left( \inf_{\boldsymbol{x} \in A \cap K^c} |f(\boldsymbol{x})|^p \right) \cdot \mu(A \cap K^c)$$

where $\mu(\cdot)$ denotes the Lebesgue measure on $\mathbb{R}^d$. Since $K$ is compact and $A$ is unbounded, $\mu(A \cap K^c) = \infty$ conclude that $\inf_{\boldsymbol{x} \in A \cap K^c} |f(\boldsymbol{x})| = 0$ for $\boldsymbol{x} \in A \cap K^c$. Since $f(\boldsymbol{x}) = \sum_{i=1}^{k_A} v_i(\boldsymbol{w}_i^\top \boldsymbol{x} + b_i) + b_0$ is linear on $A$, we have

$$\sum_{i=1}^{k_A} v_i \boldsymbol{w}_i = \boldsymbol{0} \qquad \text{and} \qquad b_0 + \sum_{i=1}^{k_A} v_i b_i = 0. \tag{34}$$

Now consider the adjacent unbounded partition $B$ such that $f$ is linear in $B$ and $B \cap K^c$ is also unbounded (such partition $B$ can be chosen by a linear partition of $f$ in $A^c \cap K^c$). Through the exactly same arguments, we obtain the similar conclusion with (34) on $B$.

$$\sum_{i=1}^{k_B} v_i \boldsymbol{w}_i = 0 \qquad \text{and} \qquad b_0 + \sum_{i=1}^{k_B} v_i b_i = 0. \tag{35}$$

However, since $B$ is the adjacent partition of $A$, exactly one neuron (call $v', \boldsymbol{w}'$) is either activated or deactivated in $B$. Comparing (34) and (35), we get either $v' = 0$ or $\boldsymbol{w}' = \boldsymbol{0}$, which contradicts to the minimality assumption. This completes the proof. $\qquad \square$

**Lemma C.3.** *Let $0 \leq m \leq d$ be integers, and $\Delta^m$ be an $m$-simplex in $\mathbb{R}^d$. For a given $\varepsilon > 0$, there exists a two-layer ReLU network $\mathcal{T} : \mathbb{R}^d \to \mathbb{R}$ with the architecture $d \xrightarrow{\sigma} (d+1) \to 1$ such that*

$$\begin{aligned} \mathcal{T}(\boldsymbol{x}) &= 1 & &\text{if } \boldsymbol{x} \in \Delta^m, \\ \mathcal{T}(\boldsymbol{x}) &\leq 1 & &\text{if } \boldsymbol{x} \in \mathcal{B}_\varepsilon(\Delta^m), \\ \mathcal{T}(\boldsymbol{x}) &< 0 & &\text{if } \boldsymbol{x} \notin \mathcal{B}_\varepsilon(\Delta^m). \end{aligned}$$

*Furthermore, the minimal width of such two-layer ReLU networks with the architecture $d \xrightarrow{\sigma} d_1 \to 1$ is exactly $d_1 = d + 1$.*

*Proof.* We prove the existence part first. For the given $m$-simplex $\Delta^m$, pick $(d - m)$ distinct points in $\mathcal{B}_\varepsilon(\Delta^m)$. By connecting all these points with the points of $\Delta^m$, we obtain a $d$-simplex contained in $\mathcal{B}_\varepsilon(\Delta^m)$, which is a convex polytope. By Proposition 3.2, there exists a neural network $\mathcal{T} : \mathbb{R}^d \to \mathbb{R}$ with the architecture $d \xrightarrow{\sigma} d_1 \to 1$ that satisfies the desired properties.

Now, we prove the minimality part. For every $\varepsilon > 0$, suppose there exists a two-layer ReLU network $\mathcal{T}(\boldsymbol{x}) := \sum_{i=1}^{d_1} v_i \sigma(\boldsymbol{w}_i^\top \boldsymbol{x} + b_i) + v_0$ with $d_1 \leq d$ such that $\mathcal{T}(\boldsymbol{x}) = 1$ for $\boldsymbol{x} \in \Delta^m$ and $\mathcal{T}(\boldsymbol{x}) < 0$ for $\boldsymbol{x} \notin \mathcal{B}_\varepsilon(\Delta^m)$. First, we claim that the set of weight vectors $\{\boldsymbol{w}_1, \cdots, \boldsymbol{w}_{d_1}\}$ spans $\mathbb{R}^d$. If the set cannot span $\mathbb{R}^d$, then there exists a nonzero vector $\boldsymbol{u} \in \mathbb{R}^d - \text{span} < \boldsymbol{w}_1, \cdots, \boldsymbol{w}_{d_1} >$. Then, from $\mathcal{T}(\boldsymbol{x}) = 1$ for $\boldsymbol{x} \in \Delta^m$, we get

$$\begin{aligned} \mathcal{T}(\boldsymbol{x} + t\boldsymbol{u}) &= \sum_{i=1}^{d_1} v_i \sigma(\boldsymbol{w}_i^\top (\boldsymbol{x} + t\boldsymbol{u}) + b_i) + v_0 \\ &= \sum_{i=1}^{d_1} v_i \sigma(\boldsymbol{w}_i^\top \boldsymbol{x} + b_i) + v_0 \\ &= \mathcal{T}(\boldsymbol{x}) \\ &= 1 \end{aligned}$$

for any $t \in \mathbb{R}$. This contradicts to the condition $\mathcal{T}(\boldsymbol{x}) < 0$ for $\boldsymbol{x} \notin \mathcal{B}_\varepsilon(\Delta^m)$. Therefore, the set of weight vectors must span $\mathbb{R}^d$.

From the above claim, we further deduce that $d_1 \geq d$. Since we start with the assumption $d_1 \leq d$, thus $d_1 = d$. Then, we conclude that the set of weight vectors $\{\boldsymbol{w}_1, \cdots, \boldsymbol{w}_{d_1}\}$ is a basis of $\mathbb{R}^d$. Now, we focus on the sign of $v_0$. Suppose $v_0 \geq 0$. Define

$$A := \bigcap_{i=1}^{d_1} \{\boldsymbol{x} \mid \boldsymbol{w}_i^\top \boldsymbol{x} + b_i < 0\},$$

which is an unbounded set since the set $\{\boldsymbol{w}_i\}$ is linearly independent. Then for $\boldsymbol{x} \in A$, we get $\mathcal{T}(\boldsymbol{x}) = v_0 \geq 0$. This contradicts to the assumption $\mathcal{T}(\boldsymbol{x}) < 0$ for all $\boldsymbol{x} \notin \mathcal{B}_\varepsilon(\Delta^m)$. Therefore, $v_0 < 0$.

Lastly, we consider the sign of $v_i$. Since $\mathcal{T}(\boldsymbol{x}) = 1 > 0$ for $\boldsymbol{x} \in \Delta^m$ and $v_0 < 0$, there exists some positive $v_i > 0$, say, $v_1 > 0$. Similar to the above argument, we define

$$B := \left\{ \boldsymbol{x} \mid v_1 \boldsymbol{w}_1^\top \boldsymbol{x} + b_1 + v_0 > 0 \right\} \bigcap_{i=2}^{d_1} \left\{ \boldsymbol{x} \mid \boldsymbol{w}_i^\top \boldsymbol{x} + b_i < 0 \right\},$$

which is also nonempty and unbounded. Then, for $\boldsymbol{x} \in B$, we have

$$\begin{aligned}
\mathcal{T}(\boldsymbol{x}) &= \sum_{i=1}^{d_1} v_i \sigma(\boldsymbol{w}_i^\top \boldsymbol{x} + b_i) + v_0 \\
&= v_1 \boldsymbol{w}_1^\top \boldsymbol{x} + b_1 + v_0 \\
&> 0.
\end{aligned}$$

Since $B$ is unbounded, this implies that $\mathcal{T}(\boldsymbol{x}) > 0$ over the unbounded subset in $\mathbb{R}^d$, which contradicts to the condition $\mathcal{T}(\boldsymbol{x}) < 0$ for all $\boldsymbol{x} \notin \mathcal{B}_\varepsilon(\Delta^m)$. This completes the whole proof, which shows that the minimum width of two-layer ReLU network is exactly $d + 1$. □

**Proposition C.4.** *Let $\mathcal{X} \subset \mathbb{R}^d$ be a topological space and $\mathcal{A}$ be a neural network architecture that is a feasible architecture on $\mathcal{X}$. Then, there exists a topological space $\mathcal{X}'$ which is homeomorphic to $\mathcal{X}$, but $\mathcal{A}$ is not a feasible architecture on $\mathcal{X}'$.*

*Proof.* We use the similar technique introduced in Telgarsky (2015). Before we start, recall that a network $\mathcal{N}$ with the architecture $\mathcal{A}$ is a piecewise linear function on $\mathbb{R}^d$. Thus $\mathbb{R}^d$ can be partitioned into finitely many regions, where $\mathcal{N}$ is linear on each region. Let $M$ be the maximum number of such regions, that networks with the architecture $\mathcal{A}$ can partition. I.e., any network with the architecture $\mathcal{A}$ has linear regions at most $M$ partitions in $\mathbb{R}^d$.

Now, we consider a contractible topological space $\mathcal{Y}$ which has zig-zag shape as described in Figure 9(b), where the number of sawtooths is greater than $M + 2$. We define another topological space $\mathcal{X}' := \mathcal{X} \# \mathcal{Y}$, where $\#$ denotes the connected sum. Note that we can glue $\mathcal{Y}$ to $\mathcal{X}$ preserving the number of sawtooths in $\mathcal{Y}$, because $\mathcal{X}$ is bounded. Then $\mathcal{X}'$ is homeomorphic to $\mathcal{X}$ since $\mathcal{Y}$ is contractible.

Finally, we prove the proposition using contradiction. Suppose there exists a deep ReLU network $\mathcal{N}'$ with the same architecture $\mathcal{A}$, which can approximate $\mathbb{1}_{\{\mathcal{X}'\}}$ under the given error bound $\varepsilon > 0$. Then, by the $\mathcal{Y}$ part in $\mathcal{X}'$, there exists a straight line $\ell$ that intersects $\mathcal{X}'$ more than $M + 3$ times. Therefore, to approximate $\mathbb{1}_{\{\mathcal{X}'\}}$ sufficiently close, $\mathcal{N}'$ must have at least $M + 1$ linear regions on $\ell$. However, $\mathcal{N}'$ can have at most $M$ linear regions in $\mathbb{R}^d$ from the definition of $M$. This contradiction completes the proof. □

**Theorem C.5.** *Let $d_x, d_y \in \mathbb{N}$ and $p \geq 1$. Then, the set of three-layer ReLU networks is dense in $L^p(\mathbb{R}^{d_x}, [0,1]^{d_y})$. Furthermore, let $f : \mathbb{R}^{d_x} \to [0,1]^{d_y}$ be a compactly supported function whose Lipschitz constant is $L$. Then, for any $\varepsilon > 0$, there exists a three-layer ReLU network $\mathcal{N}$ with the architecture*

$$d_x \overset{\sigma}{\to} (2n d_x d_y) \overset{\sigma}{\to} (n d_y) \to d_y$$

*such that $\|\mathcal{N} - f\|_{L^p(\mathbb{R}^{d_x})} < \varepsilon$. Here, $n = \varepsilon^{-d_x} \left(1 + (\sqrt{d_x} L)^p\right)^{d_x/p} = O(\varepsilon^{-d_x})$.*

*Proof.* Fist we recall a result in real analysis: the set of compactly supported continuous functions is dense in $L^p(\mathbb{R}^{d_x})$ for $p \geq 1$ (Rudin et al., 1976, Theorem 3.14). Therefore, it is enough to prove the second statement; which claims that any compactly supported Lipschitz function can be universally approximated by three-layer ReLU networks.

We consider $d_y = 1$ case first. Let $f \in \mathbb{R}^{d_x} \to [0,1]$ be Lipschitz, and let $L$ be its Lipschitz constant. Without loss of generality, suppose the support of $f$ is contained in $[0,1]^{d_x}$. Let $\delta > 0$ be the small

number which will be determined. Now we partition $[0, 1]^{d_x}$ by regular $d_x$-dimensional cubes with length $\delta$. Now, consider estimating the definite integral uses a Riemann sum over cubes. The total number of cubes are $n := (\frac{1}{\delta})^{d_x}$, and we number these cubes by $C_1, C_2, \cdots, C_n$. For each cube $C_i$, by Proposition 3.2, we can define a two-layer ReLU network $\mathcal{T}_i$ with the architecture $d_x \xrightarrow{\sigma} 2d_x \to 1$ such that $\mathcal{T}_i(\boldsymbol{x}) = 1$ in $C_i$ and $\mathcal{T}_i(\boldsymbol{x}) = 0$ for $\boldsymbol{x} \notin B_r(C_i)$ with $r := \frac{1}{2d_x} \frac{\delta^{p+1}}{1+\delta^p}$. Then for any $\boldsymbol{x}_i \in C_i$, we get

$$
\begin{aligned}
\int_{B_r(C_i)} |f - f(\boldsymbol{x}_i)\mathcal{T}_i|^p \, d\mu &= \int_{C_i} |f - f(\boldsymbol{x}_i)\mathcal{T}_i|^p \, d\mu + \int_{B_r(C_i)\setminus C_i} |f - f(\boldsymbol{x}_i)\mathcal{T}_i|^p \, d\mu \\
&\leq \int_{C_i} (\sqrt{d_x}L\delta)^p \, d\mu + \int_{B_r(C_i)\setminus C_i} 1^p \, d\mu \\
&\leq (\sqrt{d_x}L\delta)^p \cdot \delta^{d_x} + \left[(\delta + 2r)^{d_x} - \delta^{d_x}\right] \\
&= (\sqrt{d_x}L)^p \cdot \delta^{d_x+p} + \left[\left(1 + \frac{2r}{\delta}\right)^{d_x} - 1\right]\delta^{d_x} \\
&< \left[(\sqrt{d_x}L)^p + 1\right]\delta^{d_x+p}.
\end{aligned}
$$

Note that we use two inequalities, $|f(\boldsymbol{x}) - f(\boldsymbol{x}_i)| \leq L\sqrt{d_x}\delta$ for $\boldsymbol{x} \in C_i$ and $(1+a)^k < \frac{1}{1-ak}$ for $0 < a < \frac{1}{k}$. Then, the above equation implies the $L^p$ distance between $f$ and $f(\boldsymbol{x}_i)\mathcal{T}_i$ in $C_i$ is bounded by the above value. Now we define a three-layer neural network $\mathcal{N}$ by

$$
\mathcal{N}(\boldsymbol{x}) := \sum_{i=1}^{n} f(\boldsymbol{x}_i)\mathcal{T}_i(\boldsymbol{x}),
$$

which is a Riemann sum over the $n$ cubes partitions. Then $\mathcal{N}$ has the architecture $d_x \xrightarrow{\sigma} (2nd_x) \xrightarrow{\sigma} n \to 1$ and satisfies

$$
\begin{aligned}
\int_{\mathbb{R}^{d_x}} |f - \mathcal{N}|^p d\mu &= \int_{B_r([0,1]^{d_x})} |f - \mathcal{N}|^p \, d\mu \\
&< \sum_{i=1}^{n} \int_{B_r(C_i)} |f - f(\boldsymbol{x}_i)\mathcal{T}_i|^p \, d\mu \\
&\leq \left[(\sqrt{d_x}L)^p + 1\right]n\delta^{d_x+p}. \\
&= \left[(\sqrt{d_x}L)^p + 1\right]\delta^p.
\end{aligned}
$$

Therefore, take $\delta < \varepsilon(1 + (\sqrt{d_x}L)^p)^{-\frac{1}{p}}$ for given $\varepsilon$, we conclude that $\|f - \mathcal{N}\|_{L^p([0,1]^{d_x})} < \varepsilon$. From this choice of $\delta$, we get

$$
n = \delta^{-d_x} > \varepsilon^{-d_x}\left(1 + (\sqrt{d_x}L)^p\right)^{d_x/p} = O(\varepsilon^{-d_x}).
$$

If $d_y > 1$, we can obtain the desired network by concatenating $d_y$ networks, thus the architecture is

$$
d_x \xrightarrow{\sigma} (2nd_xd_y) \xrightarrow{\sigma} (nd_y) \to d_y.
$$

$\square$

**Proposition C.6** (Theorem 2.1 in Du et al. (2018), two-layer version). *Let $\mathcal{N}(\boldsymbol{x}) := v_0 + \sum_{k=1}^{l} v_k\sigma(\boldsymbol{w}_k^\top \boldsymbol{x} + b_k)$ be a two-layer ReLU network, and $L = \frac{1}{n}\sum_{i=1}^{n}\ell(\mathcal{N}(\boldsymbol{x}_i), y_i)$ be the loss function. Then, on the gradient flow, for all $k \in [l]$, the quantity*

$$
v_k^2 - \|\boldsymbol{w}_k\|^2 - b_k^2 \tag{36}
$$

*is invariant.*

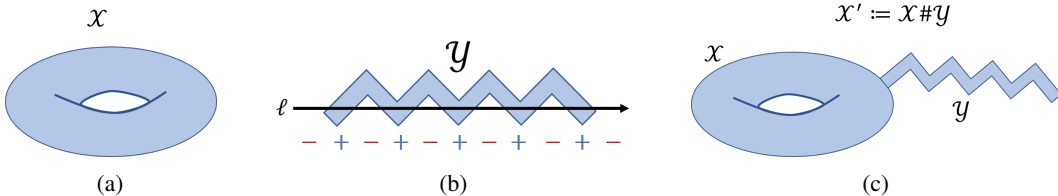

Figure 9: Proof of Proposition C.4. (a) $\mathcal{X}$ is a given topological space, and $\mathcal{A}$ is a feasible architecture on $\mathcal{X}$. (b) $\mathcal{Y}$ is a zig-zag shaped long band, which is a contractible space. There exists a straight line $\ell$ such that $\mathcal{Y}$ and $\ell$ has sufficiently many intersection points, so that $\mathcal{A}$ cannot approximate $\mathcal{Y}$. (c) $\mathcal{X}'$ is the connected sum of $\mathcal{X}$ and $\mathcal{Y}$, which is homeomorphic with $\mathcal{X}$. However, $\mathcal{A}$ is not a feasible architecture on $\mathcal{X}'$.

*Proof.* The proof is written in Du et al. (2018), and we provide here for completeness. The gradient of each component is computed by

$$\frac{\partial L}{\partial v_k} = \frac{1}{n}\sum_{i=1}^{n}\frac{\partial \ell}{\partial \mathcal{N}(\boldsymbol{x}_i)} \cdot \sigma(\boldsymbol{w}_k^\top \boldsymbol{x}_i + b_k),$$

$$\frac{\partial L}{\partial \boldsymbol{w}_k} = \frac{1}{n}\sum_{i=1}^{n}\frac{\partial \ell}{\partial \mathcal{N}(\boldsymbol{x}_i)} \cdot v_k \mathbb{1}_{\{\boldsymbol{w}_k^\top \boldsymbol{x}_i + b_k > 0\}}\boldsymbol{x}_i,$$

$$\frac{\partial L}{\partial b_k} = \frac{1}{n}\sum_{i=1}^{n}\frac{\partial \ell}{\partial \mathcal{N}(\boldsymbol{x}_i)} \cdot v_k \mathbb{1}_{\{\boldsymbol{w}_k^\top \boldsymbol{x}_i + b_k > 0\}}.$$

Then, it is easy to check that

$$v_k\frac{\partial L}{\partial v_k} = \boldsymbol{w}_k^\top \left(\frac{\partial L}{\partial \boldsymbol{w}_k}\right) + b_k \cdot \frac{\partial L}{\partial v_k}.$$

Now, we differentiate (36). It gives

$$\begin{aligned}
\frac{d}{dt}(v_k^2 - \|\boldsymbol{w}_k\|^2 - b_k^2) &= 2v_k\frac{dv_k}{dt} - 2\boldsymbol{w}_k^\top\left(\frac{d\boldsymbol{w}_k}{dt}\right) - 2b_k\frac{db_k}{dt} \\
&= 2\left(-v_k\frac{\partial L}{\partial v_k} + \boldsymbol{w}_k^\top\left(\frac{\partial L}{\partial \boldsymbol{w}_k}\right) + b_k\frac{\partial L}{\partial b_k}\right) \\
&= 0
\end{aligned}$$

for all $t$. Therefore, (36) is constant. $\qquad\square$

## D    FURTHER EXPERIMENTAL RESULTS

In this section, we present additional experimental results that specifically investigate the effect of initialization. We utilize the two datasets $\mathcal{X}_1$ and $\mathcal{X}_2$ described in Figure 4(a) and (d). Before showcasing the result, it is woth mentioning that the random initialization strategy often leads to the dying ReLU initialization issue (Lu et al., 2019), implying that the network has a constant output at initialization. Therefore, we excluded such examples and exhibit some cases where gradient descent converges. We experimented with multiple random initializations and manual initializations on neural networks with the given architecture, and the convergence results are shown in Figure 10.

Generally, reaching the global minima (near zero loss) with gradient descent is challenging, underscoring the importance of the initialization condition (9). It is anticipated that the probability of successful training increases with wider networks, as it offers more choices for selecting hyperplanes and polytopes among the neurons. Additionally, it's worth noting that the network parameters reaching the global minima are not unique, as illustrated in the second row last column of Figure 10. We leave these intriguing branches of research for future work.

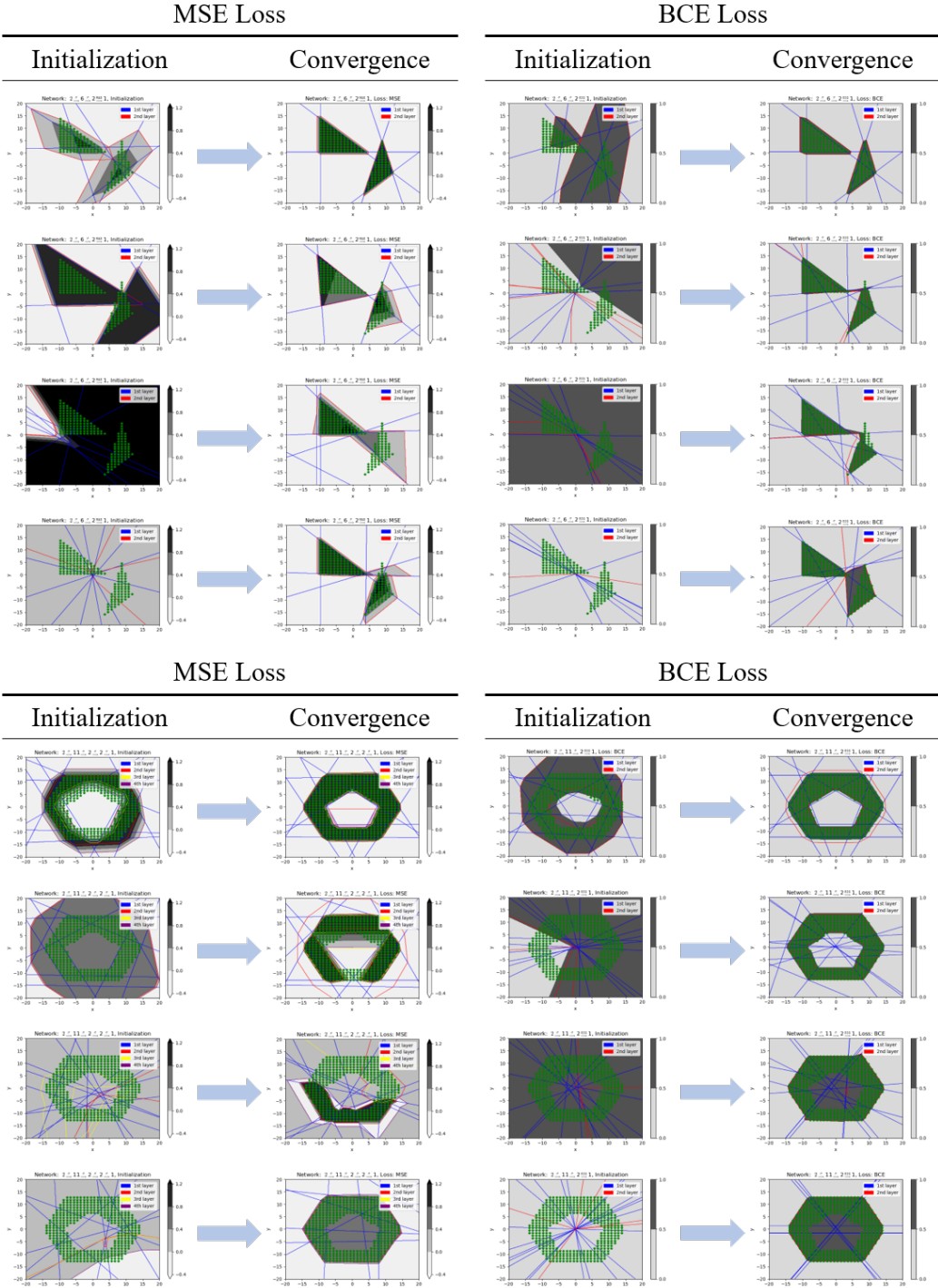

Figure 10: Additional convergence results with MSE and BCE loss functions. The result shows that gradient descent method converges to the global minimum under Assumption 4.2, as Theorem 4.3 provided. However, it may converge to a local minima when the above assumption is unsatisfied, which asserts the importance of initialization conditions.

**Detailed setup of experiments in Figure 4.** All experiments were executed using Pytorch on a GeForce GTX 1080 Ti. The training dataset was generated from $40 \times 40 = 1600$ lattice points in the $[-20, 20] \times [-20, 20] \subset \mathbb{R}^2$ range, where each point was labeled '1' if the point was in $\mathcal{X}_i$, and

'0' otherwise. The optimization method employed was full-batch gradient descent, with learning rates set at 0.005 for $\mathcal{X}_1$ (two triangles) and 0.001 for $\mathcal{X}_2$ (a hexagon with a pentagon hole). The total number of epochs was flexibly determined for each experiment, ensuring a sufficient number to achieve convergence.

## E  FINDING A POLYTOPE-BASIS COVER FOR REAL DATASET

Before delving into our results, we introduce a slight extension to the traditional notion of a convex polytope. In this extension, a convex polytope is regarded as an intersection of hyperspaces, allowing for the possibility of being unbounded.

**Definition E.1.** *A set $C \subset \mathbb{R}^d$ is called a* convex polytope with $l$ faces *if there exist $\boldsymbol{w}_k \in \mathbb{R}^d$ and $b_k \in \mathbb{R}$ for $k \in [l]$ such that $C = \bigcap\limits_{k=1}^{l} \{\boldsymbol{x} \in \mathbb{R}^d \mid \boldsymbol{w}_k^\top \boldsymbol{x} + b_k \leq 0\}$.*

According to this definition, a convex polytope may have some unbounded sides. It is crucial to emphasize that all the theoretical results presented in the main body of the paper remain applicable under this generalized definition.

Now, we propose the method to probe the geometry of the given dataset by training a two-layer ReLU network. Drawing inspiration from the constructive proof of Proposition 3.2, and the characterization of decision boundaries depicted in Vallin et al. (2023), we leverage the convexity exhibited by the two-layer ReLU network when the second layer weights $v_k$ have the same sign. This is formally demonstrated in the following proposition. We use this proposition to find a polytope-basis cover in real datasets.

*Proof of Proposition 3.7.* First, we prove that $\mathcal{N}$ is a convex function, since $\sigma$ is convex and the sum of convex functions is again convex (we use the positivity of $v_k$'s here). Therefore, the region $R := \{\boldsymbol{x} \mid \mathcal{N}(\boldsymbol{x}) < 0\}$ is convex. Further, since all $v_k > 0$, $\boldsymbol{x} \in S$ if and only if $\boldsymbol{w}_k^\top \boldsymbol{x} + b_k < 0$ for all $k \in [l]$. Then, $S$ is a convex polytope with $l$ faces by Definition E.1. The second statement is induced from the fact that the minimum of convex functions is convex. $\square$

One challenge in applying this proposition arises from the fact that in most cases, a trained neural network may have both positive and negative values for $v_k$'s. To ensure that all $v_k$ are positive, we leverage the balancedness property (Du et al. (2018, Theorem 2.1)), which is written in Proposition C.6 for completeness. It states that the quantity $v_k^2 - \|\boldsymbol{w}_k\|^2 - b_k^2$ remains invariant on the gradient flow, for all $k \in [l]$. Exploiting this identity, we initialize $v_k > (\|\boldsymbol{w}_k\|^2 + b_k^2)^{1/2}$ to ensure that all $v_k$ remain positive throughout the training process.

For our experiments, we employ three datasets: MNIST, Fashion-MNIST, and CIFAR10. We consider the binary classification distinguishing one class from all other classes, to obtain a polytope-basis cover of the class. Furthermore, it is important to highlight that in Proposition 3.7, the region classified as positive ($\boldsymbol{x} \mid N(\boldsymbol{x}) > 0$) may not exhibit convexity. Consequently, for each class, we conducted training twice using different labels, 0 and 1, to assess the convexity of the complement of each class. We represent the complement of a class as $\{\text{class}\}^c$. We trained a two-layer ReLU network (1) with increasing the width, to get 99.99% accuracy on the training set.

Our empirical results are summarized in Table 1. Each column in the table corresponds to a class in the dataset, where row presents the type of the class. The values in the table denotes the number of polytopes and their faces. Specifically, we use the notation $a + b$ to denote two polytopes with $a$ and $b$ faces, respectively. For exmple, the value in the first row and the first column shows that the $\{0\}$ class images in MNIST dataset can be distinguished with other classes, by a single convex polytope with 20 faces. On the other hand, the second row in the first column shows that the complement of the class, namely $\{0\}^c := \{1,2,3,4,5,6,7,8,9\}$, can be separated from $\{0\}$ by a convex polytope with only four faces, as illustrated in Figure 11(a). In general, our result shows that the complement of each class has more simple polytope cover, than the class itself.

In the case of Fashion-MNIST, we encountered challenges in obtaining a single convex polytope cover for certain classes. For these classes, we conducted additional training of another network to acquire more polytopes, aiming to cover the misclassified data points. Given the small number

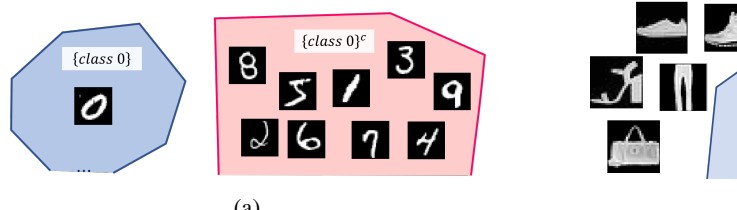

(a)                                                    (b)

Figure 11: Illustration of a polytope-basis cover in the real datasets. (a) Each class of MNIST can be separated by a single convex polytope with less than 40 faces. Furthermore, the complement of each class can be distinguished more simply - by a polytope with less than 8 faces. (b) Some classes in Fashion-MNIST or CIFAR10 are covered by a union or difference of at most three polytopes.

| | | class number | | | | | | | | | |
|---|---|---|---|---|---|---|---|---|---|---|---|
| | | 0 | 1 | 2 | 3 | 4 | 5 | 6 | 7 | 8 | 9 |
| MNIST | {class} | 20 | 20 | 40 | 30 | 20 | 30 | 20 | 40 | 40 | 30 |
| | {class}$^c$ | 4 | 7 | 4 | 7 | 4 | 4 | 3 | 5 | 7 | 7 |
| Fashion-MNIST | {class} | 20+10 | 20 | 20+20 | 30+20 | 20+35 | 10+5 | 30+25 | 10+10 | 30 | 10+5 |
| | {class}$^c$ | 10 | 20 | 40 | 10 | 40 | 10 | 40 | 10 | 10 | 10 |
| CIFAR10 | {class} | 20+10 | 20 | 30+20+5 | 30+10 | 30+10+5 | 30+10+5 | 30+10 | 30+10+10 | 30+10 | 30+10 |
| | {class}$^c$ | 30+10 | 20 | 30+30 | 30+10 | 40+10 | 20+5 | 30 | 10+10 | 30+10 | 20 |

Table 1: The number of polytopes and their faces of a polytope-basis cover of each class in MNIST, Fashion-MNIST, and CIFAR10 datasets. The result show that each class in these datasets can be separated by at most three polytopes with less than 40 faces.

of misclassified data points, we opted for a relatively small width for the second polytope. For instance, the {0} class in Fashion-MNIST can be covered by the difference of two convex polytopes, constituting a polytope-basis cover for the class. Figure 11(b) illustrates the case where the {0} class (T-shirt images) is separated by the difference of two polytopes. However, each complement set can be covered by a single convex polytope.

In the CIFAR10 dataset, we observed an increase in the number of polytopes, aligning with the intuition that CIFAR10 has a more complex geometric structure compared to MNIST or Fashion-MNIST. Despite this complexity, it is able to find a polytope-basis cover within three polytopes, each with fewer than 50 faces.

In summary, our empirical verification demonstrates that each class in the MNIST, Fashion-MNIST, and CIFAR10 datasets has a polytope-basis cover with at most three convex polytopes, each having fewer than 40 faces. This observation highlights the simple geometry inherent in these real-world datasets. Additionally, our results indicate that the complement of each class tends to have a simpler polytope-basis cover than the class itself.

Lastly, it's important to note that this doesn't imply the convexity of the data manifold. Figure 12 illustrates that although each class can be separated by a convex polytope $C$, the data manifold itself may not exhibit convexity.

**Detail setup of experiments.** We employed the BCE loss (7) to train a two-layer network. Moreover, we set $v_0$ in (1) to a fixed value of 1, treating it as a non-learnable parameter. The training process was halted once 99.99% of the training data points were contained within the polytope, as confirmed by checking the activation of ReLU neurons during training. When it was failed to find a single polytope cover for some class, we trained another network to find the second polytope distinguishing misclassified data from all other data points in different classses, and so on. As a result, we could find at most three polytopes to distiguish each class. We normalized all datasets by their mean and standard deviations.

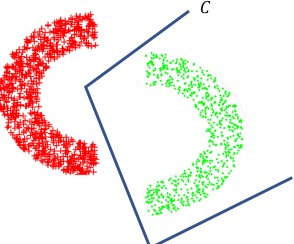

Figure 12: The existence of convex polytope cover does not imply the convexity of the dataset manifold.

