# OpenReview forum: "Data geometry and topology dependent bounds on network widths in deep ReLU networks"
_ICLR.cc/2024/Conference — Submitted to ICLR 2024_

### Official Review · Reviewer_J9Hi · 2023-10-31

**Soundness:** 2 fair
**Presentation:** 3 good
**Contribution:** 2 fair
**Rating:** 3
**Confidence:** 3

**Summary:**

Building on the observation that ReLU networks have piecewise linear/polytope decision boundaries, this paper theoretically analyzes ReLU networks for data which is a (convex) polytope or may be approximated by unions or differences of polytope. Using novel terminology, bounds on the width of ReLU networks for such polytope data are given. It is claimed these bounds are the first such bounds in terms of topological data structure. Numerical experiments are given which are claimed to verify the theory.

**Strengths:**

Analyzing learning in terms of data geometry/topology is an interesting problem. The authors make a serious effort, and I appreciate their enthusiasm.

**Weaknesses:**

The relationship of novel terminologies used in this paper to other ideas learning theory is unclear. For instance what is the relationship between the margin of a "feasible architecture" on a manifold and the generalization error of that architecture on that manifold?

I noted Proposition C.1 where the author claims that "feasible architecture" is a sufficient condition for universal approximation. I didn't find the proof convincing. You assume the existence of function $f_\delta$ which fits the indicator of $\mathcal{X}$. Then a neural network is defined to be equal to this $f_\delta$. How do you know there is such a neural network? Universal approximation would guarantee that, but that's what you're trying to prove.

The novel term "feasible architecture" is doing a lot of work in this theory. It seems to hide the question of how well an architecture actually fits a dataset. It is clear that ReLU networks can fit polytopes exactly, but it is not clear how well they can fit arbitrary manifolds.

Likewise for "polytope-basis cover". What guarantees are there for finding a tight covers for a given manifold?

Missing some discussion of prior related work on relating topological characteristics to generalization theory, e.g. [0]

Practical applicability of results is unclear. How do you actually find a polytope-basis cover?

Many strong claims are made in this paper, and after reading the paper it's not altogether clear to me they are substantiated.

Proposition 3.5 seems trivial. A very wiggly (but non-intersecting) decision boundary is homeomorphic to a linear decision boundary. The former cannot be solved by a linear classifier, but the later can be.

[0] Intrinsic Dimension, Persistent Homology and Generalization in Neural Networks - Birdal et al. (Neurips 2021)
https://proceedings.neurips.cc/paper/2021/hash/35a12c43227f217207d4e06ffefe39d3-Abstract.html

**Questions:**

Is topological space really the right level of generality for this paper? Why not just consider manifolds? The former is substantially more general, and it appears you only consider subsets of $\mathbb{R}^n$.

Can you analytically compute any explicit non-trivial examples of "feasible architecture" and "polytope-basis cover"?

Why should we believe polytope-basis covers are good approximations of manifolds? Can you prove it? Can you actually find them in practice for realistic data?

Can you clarify the relationship of your novel terms to other terms in learning theory?

Can you provide references for your proof techniques?

---

> ### Author Response · Authors · 2023-11-20
> **Reply to Reviewer J9Hi**
>
> We have summarized your concerns and questions below for a detailed response.
>
> >**W1, Q4. The relationship of novel terminologies used in this paper to other ideas learning theory is unclear. For instance, what is the relationship between the margin $\varepsilon$ of a "feasible architecture" on a manifold and the generalization error of that architecture on that manifold? Can you clarify the relationship of your novel terms to other terms in learning theory?**
>
> In our work, the margin $\varepsilon$ in Definition 3.1 signifies that if $\varepsilon$ is large, the data manifold may have a 'simple' polytope-basis cover, involving fewer polytopes or a smaller total number of faces. Consequently, the network architecture proposed in Theorem 3.4 suggests that *'the large margin cover can be approximated with a reduced number of neurons.'* In other words, increasing the filtration value $\varepsilon$ generally leads to a decreased number of neurons, establishing a connection between generalization error and network architecture from the perspective of learning theory.
>
> The insights from Theorem 3.5 can be grasped through a similar approach. In this theorem, we present a bound in terms of the dimension of the simplicial complex and the number of its faces. In TDA, the simplicial complex tends to simplify as the filtration value $\varepsilon$ increases, consequently, our bounds in Theorem 3.5 decrease. This highlights the correlation between the growth in the number of required neurons (the network width) and the increasing complexity of the dataset geometry.
>
> >**W2. Proof of Proposition C.1 is not convincing, you assume existence of function then proe NN equals to this function. How do you know NN exists? UAP guarantees but it is what you try to prove indeed?**
>
> Here we provide detailed explanations. What Proposition C.1 demonstrates is that if we can construct a constant function with its support sufficiently 'close' to $\mathcal{X}$, then we can also construct a function whose $L^p$ distance from $1_{\mathcal{X}}$ is sufficiently small.
> In simpler terms, the proposition establishes that *"constructing the indicator function"* is more difficult than *"approximating the indicator function in $L^p$ sense."* This is the reason why we solved the first problem.
> In the revised paper, we have added some sentences to provide a more detailed understanding. Refer to the first paragraph on page 3 in the revised manuscript.
>
>
> >**W3, W4, W6, Q2, Q3. It is clear that ReLU networks can fit polytopes exactly, but it is not clear how well they can fit arbitrary manifolds. What guarantees are there for finding a tight cover for a given manifold? How do you actually find a polytope-basis cover?**
>
> Indeed, this is the common concern raised by all reviewers. Please refer to General Comments KR1 in the overall comment for the detailed response.
>
>
> >**W5, Q5. Missing some discussion of prior related work on relating topological characteristics to generalization theory.**
>
> We have revised the paper to have relevant comparison with prior related work. See KR2 in General comments.
>
>
> >**W7. Many strong claims are made in this paper, and after reading the paper it's not altogether clear to me they are substantiated.**
>
> Throughout the revisions, we have improved the flow of our manuscript and emphasized our contributions. We highly recommend reviewing Section 1 in the revised manuscript for a comprehensive understanding. In summary, our main contribution lies in providing bounds on network widths that depend on the geometric characteristics of the dataset, described in terms of its polytope cover.
>
>
> >**W8. Proposition 3.5 seems trivial.**
>
> It might be understood trivially. In the revised paper, we have moved this result to the appendix, and removed from our main contributions.
>
>
> >**Q1. Is topological space really the right level of generality for this paper? Why not just consider manifolds? The former is substantially more general, and it appears you only consider subsets of $\mathbb{R}^{d}$.**
>
> Thank you for highlighting a crucial aspect. As you suggested, the role of neural networks in real-world applications is not merely to distinguish a data point within $\mathbb{R}^d$, but rather to establish a decision boundary against other classes. In response to your feedback, we have slightly generalized the definition of a *'convex polytope'* in Definition E.1 in the revised paper. Then, our theoretical results still hold with this definition, where the setting becomes more practical.
>
> Furthermore, we want to emphasize that our empirical findings in Appendix E demonstrate that each class in the real datasets (MNIST, Fashion-MNIST, CIFAR10) datasets can be effectively classified by a *'convex polytope'* characterized by a simple number of faces. Therefore, our result in Theorem 3.4 can provide the feasible architecture of neural networks on these datasets.

---

> ### Author Response · Authors · 2023-11-21
> **Dear Reviewer J9Hi**
>
> As the deadline for the Reviewer-Author discussion phase is fast approaching (there is only a day left), we respectfully ask whether we have addressed your questions and concerns adequately.

---

> ### Author Response · Authors · 2023-11-22
> **Real world data via polytope-basis cover**
>
> Dear Reviewer J9Hi,
>
> This is a gentle reminder as the deadline is approaching. We have tried our best to fully address  your questions and concerns. In particular, we have included experiments on real datasets by providing a  polytope-basis cover of real-world datasets and theoretical results to support them.
>
> Given the limited time for author-reviewer discussion, we would appreciate it if you can answer timely manner whether your queries and concerns have been satisfactorily addressed.

---

> > ### Comment · Reviewer_J9Hi · 2023-11-22
> >
> > I acknowledge the author's rebuttal and have no further questions at this time.

---

> ### Author Response · Authors · 2023-11-23
> **Thanks and follow-up**
>
> Dear reviewer J9Hi,
>
> Thanks for acknowledging our rebuttal. We are pleased to hear that you do not have any more questions. Nonetheless, we do not see the score adjustment, so we would appreciate it if you could confirm your final score in regard to our revision.  If you have further questions, please let us know we are happy to engage in further discussion.

---

> > ### Author Response · Authors · 2023-11-23
> > **confirmation?**
> >
> > Dear Reviewer J9Hi,
> >
> > We would like to kindly remind the reviewer that although you said you are satisfied with our revision and have no further questions, the score has not been changed.  Your confirmation on this matter would be highly appreciated.
> >
> > For your convenience, we would like to summarize our main  improvement in regard to your original comments:
> >
> > - KR1. We have supplied empirical results on real dataset. Specifically, we figured out the polytope-basis cover for each class in MNIST, Fashion-MNIST, and CIFAR10.
> >
> > - KR2. We have added related works, and added more descriptions compared to our work.

---

### Official Review · Reviewer_17Jw · 2023-11-05

**Soundness:** 3 good
**Presentation:** 2 fair
**Contribution:** 3 good
**Rating:** 6
**Confidence:** 4

**Summary:**

The work explores relationship between the width and, in some cases, the depth, of simple fully-connected neural networks with ReLU activations, and the polytope geometry of data distribution support, in the context of classifiers. Authors describe four constructions of  two to four layers neural networks, approximating indicator functions on convex polytopes, difference of unions of polytopes, simplicial complexes, and convex polytopes with prism-shaped polytopes removed.   Some lower bounds on the width of neural networks approximating such indicator functions  are also stated.  Finally, the work verifies that the constructed networks for convex polytopes can be reached via gradient descent optimization if it is initialized from certain regions of weight space.

**Strengths:**

1) The article describes a  construction of  two layers neural networks, approximating indicator functions on convex polytopes. The article further uses it for three constructions of three to four layers neural networks, approximating indicator functions on difference of unions of polytopes, simplicial complexes, and convex polytopes with prism-shaped polytopes removed.

2) While the polytope geometry was explored in many works in the context of ReLU fully connected neural networks, the work is based on the seemingly novel idea, see Figure 5,  that  a simple two layers ReLU network can have a constant output on a convex polytope, contrary to the more standard approach with constant output on cubical sets.

2) For three out of four constructions the work states lower bounds on width of neural networks that can approximate the indicator functions.

3) The work verifies that the constructed networks for convex polytopes can be reached via gradient descent optimization if it is initialized from certain regions of weight space.

**Weaknesses:**

Principal weaknesses of the work are:

1) Lack of applications to practical real-world datasets. It is not clear how to count the numbers of polytopes, or j-dim facets in simplicial complex, necessary to approximate a given real-world dataset, so the practical application  of the results is somewhat unclear.

2) Presentation of the paper suffers from several drawbacks. The paper contributions are theoretical, but the presented in the article proofs are too sketchy. The article is intended for a wider iclr community, but because it is too sketchy in the main part, a smooth reading  even for experts is questionable, cf below.

3) Another drawback of the presentation, in the case of the lower bounds: the principal lower bound argument involves the "bent hyperplane argument"  used in several previous works. However the paper does not explain clearly what was done in this context in the previous works compared with what constitutes the paper's novelty when establishing the lower bounds.

4) The numerous allusions to topological aspects are heavily overstated. The only place, where some topological notion appears, namely Betti numbers,  is in the fourth construction (Theorem 3.7) involving the polytopes with  prism-shaped polytopes removed. However in this context the (d-i)-th Betti number is simply the number of the removed prism-shaped convex polytopes with maximum i unbounded axes, so the Betti numbers can be replaced by such simple counters in the Theorem 3.7.

5) Related work section does not mention vast literature on geometry and topology applications in analysis of data representations, eg  Kim K et al. "Pllay: Efficient topological layer based on persistent landscapes." Advances in Neural Information Processing Systems 33 (2020), Barannikov, S et al. "Manifold Topology Divergence: a Framework for Comparing Data Manifolds." Advances in Neural Information Processing Systems 34 (2021), Barannikov, S et al. "Representation Topology Divergence: A Method for Comparing Neural Network Representations." ICML (2022).

6) The verification of possibility to reach the constructed networks via gradient descent optimization is somewhat limited as it is only applicable to some initializations satisfying some additional conditions and not to others.


Below are some specific remarks:
- page 1: The figure appearing on the article first page usually serves to highlight the principal contribution of the paper, does the standard Figure 1 with very well-known XOR dataset really serve such purpose?

- page 2: "considering the given dataset as  a topological space" -> considering the support of the data distribution as a topological space?

- page 2: "We answer this question by constructing a collection of convex polytopes..." - where is it described in the paper how to construct such collection?

- page 2: "forms m-simplicial complex..., we establish  a novel topology-dependent bound" - the bound established in Theorem 3.6 concerning simplicial complexes is in terms of numbers of j-dimensional facets, which are not "topology-dependent" quantities.

- page 4:  "from the volume identity of the polytope"- what is it? a reference is necessary here.

- page 5: in Theorem 3.6 "d_1 is bounded by" refers to the presented construction of the network, or to any 2 layer network which is feasible on X, it is not clear

- page 6: "the first result on the width of neural networks in terms of topological data structure"- the result in Theorem 3.6 is in terms of numbers of j-dimensional facets, which is not "in terms of topological data structure"

- page 7: "This implies that the architecture outlined in Theorem 3.7 is dictated by the filtration parameter"- The topological space considered in Theorem 3.7 is a convex polytope with disjoint prism-shaped convex polytopes removed, it is not the Cech complex with respect to some parameter epsilon, so there is no architecture in Theorem 3.7 related to the filtration parameter.
- page 8: "which completely classifying the given dataset " - perhaps, which classifies with zero error the given dataset ?

**Questions:**

Please address the limitations related with the feasibility of approximating  real-world datasets with the difference of polytopes or the simplicial complex constructions, how to construct and count such polytopes or the numbers of j-dimensional facets for all j.

---

> ### Author Response · Authors · 2023-11-20
> **Reply to Reviewer 17Jw (1/2)**
>
> >**W1. Lack of applications to practical real-world datasets.**
>
> In the revised paper, we supplied empirical application of our results, involving seeking a polytope-basis cover of the real-world dataset.
> See General Comment KR1 in the overall comments.
>
> >**W2. Paper does not provide the detail proofs in the main body, leading to bad presentation.**
>
> Thanks for the constructive comment. We moved less important proofs to the appendix, and allocated more paragraphs to highlight our key ideas. See General Comment KR 2 in the overall comments.
>
> >**W3. What is the novelty of paper when establishing the lower bound, especially comparison between previous works considering used term "bent hyperplane argument"?**
>
> Thank you for the valuable feedback.
> We referenced the 'arrangement of bent hyperplanes' in [Hanin et al.] as the 'bent hyperplane argument' in the original manuscript to convey that we employed a similar idea to derive a lower bound on the required number of neurons. Indeed, we used this idea to explore how the decision boundary can be 'refracted' if and only if it intersects with some activation boundary of a neuron, in the proof of Proposition 3.2.
>
> However, as there are no previous results on such a lower bound for the number of neurons based on the number of faces of a polytope, we assert that our results are novel. [Hanin et al.] and [Grigsby et al.] utilized the concept of 'bent hyperplanes' to obtain an upper bound on the number of linear regions or to investigate topological features of the decision boundary, none of which are comparable to our geometric results.
>
>
>
> **References.**
>
> [Hanin et al.] Boris Hanin and David Rolnick. Deep relu networks have surprisingly few activation patterns. Advances in neural information processing systems, 32, 2019.
>
> [Grigsby et al.] J Elisenda Grigsby and Kathryn Lindsey. On transversality of bent hyperplane arrangements and the topological expressiveness of relu neural networks. SIAM Journal on Applied Algebra and Geometry, 6(2):216–242, 2022.
>
> >**W4. The numerous allusions to topological aspects are heavily overstated. It only appears in Theorem 3.7 stating Betti numbers, with heavy assumptions on the shape.**
>
> Recognizing that our results primarily pertain to geometric rather than topological aspects, we have revised our manuscript, including adjustments to the title, to emphasize the geometric perspective rather than topological analysis. See General Comment KR3 in the General comments.
>
>
> >**W5. Related work section does not mention vast literature on geometry and topology applications in analysis of data representations.**
>
> We have revised our paper with additional related work and detail comparison with our results. See General Comment KR2 in the General comments.
>
>
> >**W6. Limitations regarding specific initialization for the verification of gradient descent.**
>
> Our convergence result in Theorem 4.3 demonstrates that the viability of approaching the proposed neural network through gradient descent. We want to claim that this result holds particular significance when compared to other UAP outcomes, such as those presented by [Park et al.] or [Cai], as they solely establish the existence of such neural networks without offering a convergence guarantee.
> Furthermore, we empirically validate that gradient descent reliably converges to the envisioned network.
>
> Furthermore, we want to highlight that the initialization condition is adaptable to dataset characteristics and not restricted to specific configurations. It varies based on the dataset distribution, illustrated by the fact that the permissible range for $v_0$ in equations (9) or (10) can widen as $\rho$ decreases—a constant linked to data distribution. In simpler terms, this expansion happens when data points are predominantly located on the surface of the separating polytope. Thus, we propose that these initialization conditions offer a mathematical representation of the dataset's distribution quality, providing insight into its inherent 'niceness.'
>
>
> **References.**
>
> [Park et al.] Sejun Park, Chulhee Yun, Jaeho Lee, and Jinwoo Shin. Minimum width for universal approximation. arXiv preprint arXiv:2006.08859, 2020.
> [Cai] Cai, Yongqiang. "Achieve the minimum width of neural networks for universal approximation." arXiv preprint arXiv:2209.11395 (2022).

---

> ### Author Response · Authors · 2023-11-20
> **Reply to Reviewer 17Jw (2/2)**
>
> >**Specific remarks: Some feedback and inquiries on figures, sentences, claims, and terminologies.**
>
> Thank you for careful reading and providing beneficial comments. Below, we answer for each remark.
>
> >**● page 1: The figure appearing on the article first page usually serves to highlight the principal contribution of the paper, does the standard Figure 1 with very well-known XOR dataset really serve such purpose?**
>
> We agree with the reviewers' comment, and have changed Figure 1 to highlight our contribution. We believe this will engage readers in understanding our motivation and contributions.
>
> >**● page 2: "considering the given dataset as a topological space" -> considering the support of the data distribution as a topological space?**
>
> We considered the data distribution $\mathcal{X}$ as a topological space. In practice, it refers the manifold of each class in the dataset.
>
> >**● page 2: "We answer this question by constructing a collection of convex polytopes..." - where is it described in the paper how to construct such collection?**
>
> The sentence would be corrected to, "We answer the question when a polytope-basis cover of $\mathcal{X}$ is given." For the construction of these covers, please refer to Section 5 of the revised manuscript.
>
> >**● page 2: "forms m-simplicial complex..., we establish a novel topology-dependent bound" - the bound established in Theorem 3.6 concerning simplicial complexes is in terms of numbers of j-dimensional facets, which are not "topology-dependent" quantities.**
>
> Indeed, rigorously speaking, the dimension of a simplicial complex is not a topology-dependent quantity, despite the use of simplicial complexes to explore topological properties. What we aimed to highlight was the demonstration that the bound depends on the dimension of the simplicial complex, which may be related to the topological complexity of $\mathcal{X}$. In response to your insightful suggestion (W4 above), we have removed topological aspects in the revised paper.
>
> >**● page 4: "from the volume identity of the polytope"- what is it? a reference is necessary here.**
>
> Thank you for a good comment. It means that the volume of a convex polytope is sum of the volume of small pyramids which is a partition of the polytope, which we only described in the proof in Appendix B.1.1. We allocated more space to explain this proof idea in the main body, in the revised version.
>
> >**● page 5: in Theorem 3.6 "$d_1$ is bounded by" refers to the presented construction of the network, or to any 2 layer network which is feasible on X, it is not clear**
>
> The $d_1$ refers the width in the presented construction of the network, not any 2-layer. We have modified the sentence.
>
> >**● page 6: "the first result on the width of neural networks in terms of topological data structure"- the result in Theorem 3.6 is in terms of numbers of j-dimensional facets, which is not "in terms of topological data structure"**
>
> This is similar to one of the previous comments above. We initially considered it to be related to topological data structure, but we have removed some sentences that might be understood as an overstatement.
>
> >**● page 7: "This implies that the architecture outlined in Theorem 3.7 is dictated by the filtration parameter"- The topological space considered in Theorem 3.7 is a convex polytope with disjoint prism-shaped convex polytopes removed, it is not the Cech complex with respect to some parameter epsilon, so there is no architecture in Theorem 3.7 related to the filtration parameter.**
>
> Thank you for the significant comment. We decided to remove this part from our draft.
>
> >**● page 8: "which completely classifying the given dataset " - perhaps, which classifies with zero error the given dataset?**
>
> The sentence means the classifier with zero error on the given dataset. We have modified the sentence in the revised version.

---

> > ### Comment · Reviewer_17Jw · 2023-11-23
> >
> > I've read the authors comments and appreciate the  improvements made in the text. I'm raising my score accordingly.

---

> > > ### Author Response · Authors · 2023-11-23
> > > **Thank you for raising score.**
> > >
> > > Dear Reviewer 17Jw,
> > >
> > > We are pleased to hear that our response and revision have successfully addressed your concerns. Thank you very much for your positive response and raising the score.

---

### Official Review · Reviewer_nRVF · 2023-11-09

**Soundness:** 3 good
**Presentation:** 4 excellent
**Contribution:** 3 good
**Rating:** 8
**Confidence:** 4

**Summary:**

The paper suggests feasible shallow ReLU-induced neural network architectures that approximate indicator functions on a space having a polytope basis cover. It proposes lower and upper bounds to network widths in the process, based on Betti numbers in case the underlying space has prism-shaped convex holes. The proposed networks can also be realized based on gradient descent, by minimizing some of the commonly used loss functions.

**Strengths:**

The organization and writing are of sound quality. The theoretical framework is technically solid and the results clearly address the problem being dealt with. The supporting experiments provide empirical evidence for the findings.

**Weaknesses:**

There remain a few typographical/grammatical errors in the manuscript.

**Questions:**

1. It is often difficult for real data sets to satisfy Assumption 4.2 since the polytopes separating the clusters may not be convex. Are there any definitive modifications to the proposed convergence guarantees [Theorem 4.3] that the authors can suggest? How does the increase in the number of classes, and hence perhaps overlapping polytopes add to the complexity of the construction [Page 21]?

2. The optimality of $3$-layer ReLU networks for approximating indicators is clear from Proposition C.1 and C.2. Is it true for high-dimensional compactly supported functions ($f: R^d \to R^l$), $l>1$ in general, perhaps of some regularity in terms of smoothness? This seems crucial as there are numerous UA bounds for Lipschitz maps using ReLU feed-forward networks.

3. Can the authors comment on how well the prescriptions regarding architectures hold up against simpler real datasets? As a practitioner, it is often frustrating to witness theoretical suggestions underperforming significantly. For example, to my knowledge, there exists no consistent method of estimating Betti numbers corresponding to even simpler real data distributions, if they at all have punctured supports.

---

> ### Author Response · Authors · 2023-11-20
> **Reply to Reviewer nRVF**
>
> >**Q1. How Theorem 4.3 can be improved, for example, when the dataset is non-convexly separated or the number of classes is greater than 2?**
>
> Theorem 4.3 explains that if a two-layer ReLU network is initialized in proximity to the polytope cover, gradient descent may converge to the global minimum. This result readily extends to multiple polytopes or a polytope-basis cover.
> Specifically, consider a scenario with $m$ polytopes covering a single class and a three-layer network $\mathcal{T}$ as defined in Proposition 3.7, equation (5) in the revised paper. Assuming the $m$ subnetworks, denoted as $\mathcal{N}_i$, are initialized near polytopes, satisfying the conditions in Theorem 4.3.
> Then, we can apply Theorem 4.3 to each polytope individually.
> This is possible because $\mathcal{T}$ is the minimum of $m$ two-layer ReLU networks - it is important to note that the $\min$ operation in the network executes all gradients in backpropagation, except only one subnetwork for each input. This establishes the extension of Theorem 4.3 to a polytope-basis cover for three-layer ReLU networks.
>
> >**Q2. Is it possible to obtain some UAP results of the 3-layer ReLU networks to approximate a compactly supported functions $f:\mathbb{R}^{d_x} \rightarrow \mathbb{R}^{d_y}$, in terms of smoothness?**
>
> Thank you for a insightful question. We obtained that it is possible.
> Since we proposed a method to approximate the indicator function in $\mathbb{R}^d$ by a 3-layer ReLU network, a common idea in Lebesgue theory easily generalizes this result to approximate a compactly supported functions $f:\mathbb{R}^{d_x} \rightarrow \mathbb{R}^{d_y}$.
> We propose the result in the appendix. See Theorem C.5 in the revised paper.
> As the reviewer anticipated, the width is related with the error bound and the Lipshictz constant of $f$.
>
> Remark. Notably, (Wang et al., 2022) recently proved that 2-layer ReLU networks cannot approximate a compactly supported function in $L^p(\mathbb{R}^d)$, while 3-layer ReLU networks can. However, their proof establishes only the existence of such networks without offering any width bounds.
> Consequently, our result in Theorem C.5 represents a refinement of their findings.
> This is one example that our results on polytope can be leveraged to derive width bounds for the UAP.
>
> **Reference.**
>
> [Wang et al., 2022] Ming-Xi Wang and Yang Qu. Approximation capabilities of neural networks on unbounded domains. Neural Networks, 145:56–67, 2022.
>
>
> >**Q3. How can your results be applied to real datasets?**
>
> In the revised paper, we supplied empirical application of our results, by constructing a polytope-basis cover of the given real  dataset.
> See General Comment KR1 in the overall comments.

---

### Official Review · Reviewer_eavJ · 2023-11-09

**Soundness:** 3 good
**Presentation:** 2 fair
**Contribution:** 3 good
**Rating:** 6
**Confidence:** 3

**Summary:**

The authors study the ability of neural networks to approximate the indicator
function for $\epsilon$-blowups of convex polytopes (or more generally, the
blowup of a difference of unions of convex polytopes) as a function of the width
and depth of the neural network, versus the complexity of the polytope. The
authors present a general result that says (for example) that a two-layer
network with a number of neurons in the hidden layer growing with the number of
hyperplanes that define the convex polytope $\mathcal{X}$ is sufficient to
represent exactly the indicator function for $\mathcal{X}$ (with a corresponding
lower bound). Further results are given that quantify the width in terms of
Betti numbers or $k$-facets when $\mathcal{X}$ is a simplical complex, as well
as providing a local (initialization-dependent) theory for obtaining such
networks via global minimization of an empirical risk over the data
distribution. Low-dimensional experimental results are presented that verify
that gradient descent finds networks matching the architectural parameters
asserted as sufficient by the theory for two toy data distributions.

**Strengths:**

- The mathematical writing in the paper is clear and precise. The authors define
  relevant concepts, precisely state hypotheses, include relevant ancillary
  results in appendices with appropriate references, and present a rather robust
  characterization of the problem (in terms of sufficient architectures for
  representing indicators for convex polytopes (with holes), and results that
  specify the widths in terms of complexity parameters of these polytopes).

- The experimental results consider toy (low-dimensional) cases, but present a
  compelling verification of the conclusions of the authors' theoretical
  results.

**Weaknesses:**

- The metrics in the paper used to quantify geometric structure in the input
  data (which the theoretical results reflect in terms of the rates for the
  network widths to achieve the feasible architecture property) seem that they
  may be hard to compute in moderate dimensions (presumably, one needs to fit a
  simplical complex to data, and calculate Betti numbers from it). This means it
  may be hard to verify the theory in cases beyond the low-dimensional examples
  highlighted in experiments. I hope the authors will correct me if I am
  mistaken here, and clarify this in the revision.

- The non-technical writing in the paper (in contrast to the mathematical
  writing, highlighted above) suffers from a lack of precision in many areas. I
  would recommend rewriting the abstract to be more in line with the tone of the
  rest of the paper, tuning the first sentence of the introduction (I do not
  think this claim is universally accepted -- arguably, the ability to
  (efficiently) learn these networks is of far greater importance for
  understanding the successes of deep learning), and generally proofreading for
  typos.

- There are two relevant references that I think should be discussed in this
  context -- both are relevant to guarantees for *learning deep networks* when
  the data distribution has nontrivial geometric structure, going beyond the
  present theory on initialization-dependent or pure-representation-capacity
  results. The first is [1], which proves classification guarantees for random
  three-layer neural networks with rates that depend on the geometric structure
  of the input (measured through the Gaussian width). I think it could inform
  the presentation in the present submission to contrast with this work, as the
  way this work proves its results is by studying the way the random
  initialization induces a hyperplane arrangement conducive to separation
  (perhaps similar to the authors' analysis, for representation capacity).
  The second is [2-3], which studies sufficient settings of width and depth to
  classify pairs of one-dimensional curves (in terms of geometric properties of
  the data) with a deep ReLU network trained with gradient descent. This work
  uses very different tools from the present submission, but its motivation is
  relevant, and contrasting with this work may allow the authors to highlight
  salient advantages of their tools/framework.


[1] Dirksen, S., Genzel, M., Jacques, L., & Stollenwerk, A. (2022). The
Separation Capacity of Random Neural Networks. Journal of Machine Learning
Research: JMLR, 23(309), 1–47.

[2] Buchanan, S., Gilboa, D., & Wright, J. (2021). Deep Networks and the
Multiple Manifold Problem. International Conference on Learning Representations.
https://openreview.net/forum?id=O-6Pm_d_Q-

[3] Wang, T., Buchanan, S., Gilboa, D., & Wright, J. (2021). Deep Networks
Provably Classify Data on Curves. In M. Ranzato, A. Beygelzimer, Y. Dauphin, P.
S. Liang, & J. W. Vaughan (Eds.), Advances in Neural Information Processing
Systems (Vol. 34, pp. 28940–28953). Curran Associates, Inc.

**Questions:**

- Can the authors clarify the reason for the focus on representing indicators
  for the relevant polytopes $\mathcal{X}$ (in the kind of $L^\infty$ sense
  mandated by Definition 3.1), and the limitations of this framework for general
  problems of interest? It seems to me that it might be too "hard" of a problem
  to characterize sufficient architectural configurations to fit data
  distributions if one's end goal is a machine learning task such as
  classification (for example, generally, one could classify $\mathcal{X}$
  without exactly representing $\mathbb{1}(\mathcal{X})$). It also seems to me
  that representing $\mathbb{1}(\mathcal{X})$ may not be sufficient to solve
  general nonparametric regression tasks, i.e. to enjoy universal approximation
  of various nonparametric classes defined on $\mathcal{X}$ (please correct me
  if I am mistaken). A significant amount of work has been done in the latter
  setting, specifically on manifolds, which does not seem to have been
  discussed (e.g., [4] and many works by the same authors).

- In Section 3.1, it is not exactly explicitly defined what an "architecture" is
  (relevant to understanding $\mathcal{A}$ in Definition 3.1), but it seems from
  context that it is a fixed choice of inter-layer maps and in particular of
  hidden layer dimensions. A limitation of this definition (c.f. footnote 1)
  seems to be that in general, universal approximation of various nonparmetric
  classes can only be enjoyed with neural networks when the hidden layer
  dimension is allowed to grow -- in other words, the "architecture" involves
  only (in a sense) the computational graph of the neural network, rather than
  particulars (such as input and output dimensions) about the maps corresponding
  to "edges" in the graph. Could the authors clarify this difference, and why
  they have opted to define an "architecture" in this way?

[4] Chen, M., Jiang, H., Liao, W., & Zhao, T. (2022). Nonparametric regression on
low-dimensional manifolds using deep ReLU networks: function approximation and
statistical recovery. Information and Inference: A Journal of the IMA, iaac001.

---

> ### Author Response · Authors · 2023-11-20
> **Reply to Reviewer eavJ**
>
> >**W1. The geometric structure of the given dataset (the simplicial complex or Betti numbers) looks hard to be achieved in real dataset.**
>
> To address your concern, in the revised paper, we present a way to find a polytope-basis cover for the given real dataset. See General Comment KR1 for more details of real experimental results.
>
>
> >**W2 & W3. The abstract and certain sentences in introduction is highly recommended to be revised. Further, there are some lack of introducing related works and detail comparison.**
>
> We thank for suggesting relevant references and valuable feedback. We have revised Section 1 and 2 with additional references.
> See General Comment KR2 in the overall comments.
>
>
> >**Q1. Why did the authors choose to approximate the indicator function?**
>
> Thank you for the valuable feedback. Indeed, approximating the indicator function is generally more difficult than just classifying finite classes in the dataset. However, if a neural network $\mathcal{N}$ can effectively approximate the indicator function on a given dataset class $\mathcal{X}$, it inherently possesses the capability to classify $\mathcal{X}$ from other classes. We intended to set the goal of our paper to cover not only classification tasks, but also regression tasks. We have indeed considered the MSE loss function in Section 4 (see Theorem 4.3), and we have included a simple approximation result in Theorem C.5, requested by Reviewer nRVF. See footnote 1 on page 4 of the revised manuscript for more detail.
>
>
> >**Q2. What does the terminology `architecture' (denoted by $\mathcal{A}$) exactly mean?**
>
> In our manuscript, the terminology `architecture' refers to the structure of neural networks, specifically the depth and width of each hidden layer. We introduced a notation, $d \rightarrow d_1 \rightarrow \cdots \rightarrow d_k \rightarrow 1$, to represent a $k$-layer neural network with widths $d_1, \cdots, d_k$ in the manuscript, and we did not fix the network architecture.
>
> As you commented, the UAP result can be achieved when the the hidden layer dimension is allowed, as presented in Theorem C.5.
> More precisely, the hidden widths grows $O(\varepsilon^{-d_x})$ as the error bound  $\varepsilon$ reduces. Similarly, our other results (Theorem 3.2, 3.4, 3.5, 3.6) imply that the width increases as the number of polytopes and their faces increases. Precisely, Theorem 3.5 explicitly demonstrates how the width grows with the increasing complexity of the simplicial complex.
> We have made this part clearer in the revised manuscript (see Section 3.1 and Definition 3.1 of the revised paper).

---

> > ### Comment · Reviewer_eavJ · 2023-11-22
> > **thanks**
> >
> > Dear authors,
> >
> > Thanks for your response to my review. The new experiments are interesting. Because I believe these methodologies present an interesting alternate perspective on how to investigate these issues of deep networks and structured data relative to the mainstream, I will increase my score.

---

> > > ### Author Response · Authors · 2023-11-23
> > > **Thanks!**
> > >
> > > Dear Reviewer eavJ,
> > >
> > > Thanks for your positive comment and for raising the score. We are happy to hear that our new experiments are interesting and our methodologies present an interesting alternative perspective on how to investigate deep networks.

---

### Author Response · Authors · 2023-11-20
**General Comments**

We sincerely appreciate the thoughtful and constructive feedback provided by the reviewers. Their insightful comments have been instrumental in refining our work and enhancing its overall quality. In response to the valuable feedback, we have made several significant improvements to our manuscript. We marked by blue color for the important changes.
The Key Revisions (**KR**s) are summarized below:

>**KR1. We have supplied empirical results on real dataset. Specifically, we figured out the polytope-basis cover for each class in MNIST, Fashion-MNIST, and CIFAR10.**

Most of reviewers concerned the existence of the *polytope-basis covers* (Definition 3.3), and how they can be obtained from the real dataset. To address this concern, we introduce a method that to find such a polytope-basis cover.
For MNIST, Fashion MNIST, and CIFAR10 datasets, we indeed found the polytope-basis cover for each class. Intriguingly, our experimental results demonstrate that each class can be classified by at most three convex polytopes which having a few faces. We believe these results answer the reviewers' questions. See Section 3.3 and Appendix E in the revised paper.


>**KR2. We have added related works, and added more descriptions compared to our work.**

Responding to reviewers' suggestions about missing related work, we have revised the introduction and related work sections. We have incorporated relevant previous work and provided a thorough comparison with our results to highlight the distinct contributions of our approach (see Sections 1 and 2).


>**KR3. We have reorganized our manuscript.**

In response to the suggestions from Reviewers eavJ, nRVF, and 17Jw regarding enhancing the manuscript's fluency, we conducted a comprehensive reorganization, accompanied by thorough proofreading. In particular, we have emphasized the relationship between the geometrical features of datasets and network architecture, where some imprecise descriptions on topological explanations has been removed. We have removed the word ‘topology’ in the title, and we have refined the overall flow of the paper.

We have enhanced the rigor of certain sentences and definitions by providing more clear and detailed descriptions. Following Reviewer 17Jw's comment, the main idea of our work to emphasize is located in the main paper, and other proofs are moved to Appendix. To make it easier for readers, following the feedback of Reviewer 17Jw, we have changed some figures and rearranged existing figures.

---

### Meta-Review · Area_Chair_qNLn · 2023-12-14

**Metareview:**

The paper studies the complexity of approximating datasets with ReLU networks. The paper is based on the observation is that the ReLU networks can concisely approximate the indicator for a convex polytope with a small H description (intersection of a small number of half-spaces) since a single neuron (followed by a subsequent max operation) can approximate the indicator for a single half space. The paper develops extends this notion. To datasets which are unions of such “H-simple” polytopes and their set differences with “H-simple” polytopes. The existence side of the paper’s results pertains to shallow networks (one hidden layer for polytopes / unions and three hidden layers for differences). The paper also proves lower bounds on the complexity of deep approximations to these objects, and proves a result on the local geometry of optimization, which says that under certain initialization conditions, there exists a path of decreasing loss which links the initialization and this target approximation.

Initial reviews of the paper were split: the paper provides a novel perspective on the relationship between data geometry / topology and network complexity. The mathematics is crisply stated, and provides both upper and lower bounds on the complexity of networks for datasets that are well approximated in terms of polytopes.

Issues included
1. concerns about writing and
2. generality / applicability of the proposed approach, since it may be a-priori unclear whether a given dataset admits a simple polytope description, or whether polytope approximations capture the simplicity of other "low-complexity" data objects such as manifolds.
Reviewers raised a number of smaller issues with the paper which were largely addressed in the authors response. The response also provided experiments showing that several datasets such as CIFAR-10 and MNIST can be approximated with simple polytopes.

**Justification For Why Not Higher Score:**

The paper provides a novel perspective on approximating datasets with ReLU networks, arguing that datasets with simple polytope descriptions admit simple networks. The main reviewer concern was around the general applicability of this approach: whether it can generate prescriptions for network architectures / complexities for datasets whose structure is a-priori unknown, and whether complex datasets admit simple polytope descriptions. While the author response is a step in this direction, these experiments did not fully overcome reviewer concerns about whether the proposed approach can offer guidelines for choosing the size of networks in practice and about the generality of this "simple description" phenomenon.

**Justification For Why Not Lower Score:**

N/A

---

### Decision · Program_Chairs · 2024-01-16

Reject